# RNA Sequencing Reveals Inflammatory and Metabolic Changes in the Lung and Brain After Carbon Black and Naphthalene Whole Body Inhalation Exposure in a Rodent Model of Military Burn Pit Exposures

**DOI:** 10.3390/ijms26157238

**Published:** 2025-07-26

**Authors:** Allison M. Haaning, Brian J. Sandri, Henry L. Wyneken, William T. Goldsmith, Joshua P. Nixon, Timothy R. Nurkiewicz, Chris H. Wendt, Paul Barach, Janeen H. Trembley, Tammy A. Butterick

**Affiliations:** 1Minnesota Supercomputing Institute, University of Minnesota, Minneapolis, MN 55455, USA; haani001@umn.edu; 2Minneapolis Veterans Affairs Health Care System, Minneapolis, MN 55417, USA; brian.sandri@va.gov (B.J.S.); henry.wyneken@va.gov (H.L.W.); joshua.nixon@va.gov (J.P.N.); christine.wendt@va.gov (C.H.W.); trem0005@umn.edu (J.H.T.); 3Department of Physiology, Pharmacology, and Toxicology, West Virginia University, Morgantown, WV 26506, USA; wgoldsmi@hsc.wvu.edu (W.T.G.); tnurkiewicz@hsc.wvu.edu (T.R.N.); 4Department of Surgery, University of Minnesota, Minneapolis, MN 55455, USA; 5Center for Inhalation Toxicology, West Virginia University, Morgantown, WV 26506, USA; 6Department of Medicine, University of Minnesota, Minneapolis, MN 55455, USA; 7Department of Safety and Quality Science, College of Population Health, Thomas Jefferson University, Philadelphia, PA 19107, USA; paul.barach@gmail.com; 8Sheps Center for Health Services Research Center, University of North Carolina, Chapel Hill, NC 27599, USA; 9Department of Laboratory Medicine and Pathology, University of Minnesota, Minneapolis, MN 55455, USA; 10Masonic Cancer Center, University of Minnesota, Minneapolis, MN 55455, USA; 11Department of Food Science and Nutrition, University of Minnesota, St Paul, MN 55108, USA; 12Department of Neuroscience, University of Minnesota, Minneapolis, MN 55455, USA

**Keywords:** military burn pit exposure, military personnel, environmental exposure, respiratory diseases, lung diseases, brain diseases, inflammation, cytokines, gene expression profiling, oxidative stress, tumor necrosis factor-alpha (TNF-α), interferon-gamma (IFN-γ), air pollutants, nanoparticles, polycyclic aromatic hydrocarbons, volatile organic compounds, elastic net, gene set enrichment analysis (GSEA)

## Abstract

Military personnel deployed to Iraq and Afghanistan were exposed to emissions from open-air burn pits, where plastics, metals, and medical waste were incinerated. These exposures have been linked to deployment-related respiratory diseases (DRRD) and may also impact neurological health via the lung–brain axis. To investigate molecular mechanisms, adult male rats were exposed to filtered air, naphthalene (a representative volatile organic compound), or a combination of naphthalene and carbon black (surrogate for particulate matter; CBN) via whole-body inhalation (six hours/day, three consecutive days). Lung, brain, and plasma samples were collected 24 h after the final exposure. Pro-inflammatory biomarkers were assessed using multiplex electrochemiluminescence and western blot. Differentially expressed genes (DEGs) were identified by RNA sequencing, and elastic net modeling was used to define exposure-predictive gene signatures. CBN exposure altered inflammatory biomarkers across tissues, with activation of nuclear factor kappa B (NF-κB) signaling. In the lung, gene set enrichment revealed activated pathways related to proliferation and inflammation, while epithelial–mesenchymal transition (EMT) and oxidative phosphorylation were suppressed. In the brain, EMT, inflammation, and senescence pathways were activated, while ribosomal function and oxidative metabolism were downregulated. Elastic net modeling identified a lung gene signature predictive of CBN exposure, including Kcnq3, Tgfbr1, and Tm4sf19. These findings demonstrate that inhalation of a surrogate burn pit mixture induces inflammatory and metabolic gene expression changes in both lung and brain tissues, supporting the utility of this animal model for understanding systemic effects of airborne military toxicants and for identifying potential biomarkers relevant to DRRD and Veteran health.

## 1. Introduction

Military personnel deployed to combat zones such as Iraq and Afghanistan over the past two decades have been exposed to multiple environmental hazards, including emissions from open-air burn pits used for waste disposal [1,2,3]. Open burn pits were the primary means of waste disposal on military bases in these regions, generating complex mixtures of toxic pollutants that have been increasingly linked to respiratory diseases, cancers, systemic and central inflammation [1,4,5,6,7,8]. The usage of open burn pits, which incinerate a wide range of materials including plastics, metals, medical waste, electronics, chemicals, and petroleum products, has raised significant concerns about potential long-term occupational health effects in Veterans [9]. The Department of Defense estimates that large army base sites collectively burned up to 85,000 pounds of waste per day, with the largest burn pit at Joint Base Balad spanning over 25 acres at its peak [10].

Burn pit emissions are characterized by high concentrations of particulate matter (PM), including fine particulate matter with a diameter of 2.5 μm or less (PM_2.5_), which frequently far exceeded U.S. air pollution standards [2,11]. The emissions also contained a complex mixtures of toxic gases, such as polycyclic aromatic hydrocarbons (PAHs) and volatile organic compounds (VOCs), which have been linked to a variety of acute and chronic health issues [10,11,12,13,14]. Exposure to burn pits has been associated with deployment-related respiratory diseases (DRRD) which encompass a spectrum of debilitating pulmonary conditions, including constrictive bronchiolitis and chronic obstructive pulmonary disease [15]. Emerging research also highlights the broader systemic effects of burn pit-derived pollutants, implicating them in neuroinflammation and immune dysregulation through pathways such as the lung-brain axis [4,5,8,16,17,18]. Burn pit toxicants, including PM, VOCs, and PAHs, are linked to neuroinflammation and increased risk for neurodegenerative conditions such as Alzheimer’s disease (AD) and Parkinson’s disease (PD) [8,17,19]. One of the most thoroughly examined hypotheses concerns the adverse health effects of airborne burn pit emissions on the activation of inflammatory responses [9,16,20]. These inflammatory responses are thought to mediate the causal pathways from respiratory exposures to the development of pulmonary, cardiovascular, neurological, and malignant diseases [17,21,22,23]. It is essential to understand these mechanisms to accurately assess the long-term health risks and develop targeted interventions for affected Veterans.

To model military burn pit emissions, we exposed rats to either naphthalene (NA) alone or to a combination of carbon black nanoparticles and naphthalene (CBN). The stepwise design approach allowed us to isolate the effects of VOCs and PAHs, as represented by naphthalene, as well as their combined effects with PM_2.5_, represented by carbon black (CB). Our prior studies demonstrated that CB exposures alone can induce both systemic and central inflammation, but that model was limited by its inability to capture the complexity of burn pit toxicants [16]. The addition of naphthalene addressed this gap by incorporating key vapor-phase components common to military burn pit emissions [10,11].

Following the toxicant exposures, we conducted protein biomarker profiling, which revealed that the CBN group exhibited the most pronounced inflammatory response across lung, plasma, and brain tissues. Based on this robust protein-level activation of inflammatory pathways, we selected CBN-exposed lung and brain tissues for RNA-seq to explore gene expression changes underlying this response. We also applied predictive modeling to gene expression in the lung, the primary site of contact with airborne burn pit emissions, to identify early molecular indicators of inhalational toxicant exposure. Among all tissues analyzed, the lung exhibited the most robust transcriptional responses, making it an ideal candidate for developing a gene expression signature reflective of direct environmental insults. An elastic net model identified genes predictive of CBN exposure in lung. By delineating key genes and molecular pathways, we identified potential biomarkers of exposure and early indicators of disease progression, supporting our long-term goal of developing diagnostic and therapeutic strategies to help diagnose and treat Veterans affected by military environmental exposures.

## 2. Results

### 2.1. Whole-Body Inhalation Exposure of Rats to Sham Air, Naphthalene, or Carbon Black with Naphthalene

Rats were exposed via whole-body inhalation to either HEPA-filtered air (sham control group), NA vapor, or CBN for six hours per day over three consecutive days. For the CBN group, the aerosol mass concentration was 7.58 ± 1.55 mg/m^3^ and the naphthalene vapor concentration was maintained at 10.16 ± 0.01 parts per million (PPM). Aerosol characterization revealed a median particle diameter of 154 nm with a geometric standard deviation of 2.71, confirming that the aerosolized particles were within the respirable range for rodent inhalation studies. These exposure levels were selected to simulate realistic and environmentally relevant conditions based on those reported near military burn pit sites [24]. Naphthalene was chosen for its high volatility, well-characterized toxicological profile, and established role in respiratory tract lesions and carcinogenicity in rodent models [11,13,16,25]. The carbon black concentration represents PM levels that frequently exceed air quality standards in deployed settings. Together, the combined exposure to carbon black and naphthalene was designed to more accurately model the complex airborne pollutant mixtures generated by open-air burn pits during military operations [2,10,11,16].

### 2.2. Biomarkers of Inflammation

In lung tissue, exposure to CBN induced a broad and statistically significant inflammatory response compared with both sham and NA groups (Figure 1). IL-1β levels were markedly elevated in the CBN group (67.68 ± 8.48 pg/mL) relative to NA (54.53 ± 6.04 pg/mL) and sham controls (47.27 ± 8.92 pg/mL). Interestingly, IL-6 levels were significantly decreased in both NA (34.48 ± 13.05 pg/mL) and CBN (40.85 ± 17.37 pg/mL) groups compared with sham animals (103.6 ± 21.72 pg/mL), suggesting a selective suppression of this cytokine. IFN-γ levels increased significantly in both NA (4.96 ± 0.87 pg/mL) and CBN (5.29 ± 0.98 pg/mL) groups compared with sham (2.40 ± 0.53 pg/mL), indicating enhanced immune activation. A similar pattern was observed for IL-4, which rose in NA (0.458 ± 0.169 pg/mL) and CBN (0.514 ± 0.094 pg/mL) groups compared with sham (0.278 ± 0.095 pg/mL).

Anti-inflammatory cytokine IL-10 in lung was also significantly elevated in NA (5.10 ± 0.74 pg/mL) and CBN (5.42 ± 0.99 pg/mL) groups relative to sham (3.19 ± 1.17 pg/mL). KC/GRO (CXCL1), a chemokine involved in neutrophil recruitment, showed the most dramatic increase, reaching 153.2 ± 32.5 pg/mL in the CBN group and 69.96 ± 17.72 pg/mL in the NA group, compared with 42.86 ± 12.13 pg/mL in sham controls. IL-5 levels were also elevated, with values of 18.22 ± 2.23 pg/mL and 14.01 ± 4.06 pg/mL in the CBN and NA groups, respectively, compared with 12.30 ± 1.06 pg/mL in the sham group. IL-13 levels increased significantly in both NA (1.20 pg/mL) and CBN (1.34 pg/mL) exposures relative to sham (0.54 pg/mL). Lastly, TNF-α was significantly elevated in the CBN group (3.51 ± 0.44 pg/mL) and to a lesser extent in the NA group (2.57 ± 0.49 pg/mL), compared with sham (2.38 ± 0.45 pg/mL). Collectively, these data demonstrate that CBN exposure leads to a pronounced and multifaceted inflammatory cytokine response in the lung, with greater effects than NA alone, supporting the utility of the CBN exposure model for studying deployment-related respiratory inflammation.

In plasma, multiple pro-inflammatory cytokines were significantly elevated in the CBN-exposed group compared with sham controls, indicating systemic immune activation following inhalation exposure (Figure 2). IL-4 levels increased significantly in both NA (0.893 ± 0.302 pg/mL) and CBN (0.981 ± 0.232 pg/mL) groups compared with sham (0.043 ± 0.095 pg/mL). IL-6 concentrations were also markedly elevated, with CBN (28.11 ± 9.72 pg/mL) and NA (28.38 ± 9.72 pg/mL) groups both showing significant increases over sham (1.64 ± 9.72 pg/mL). Similarly, IL-10 levels rose substantially in the CBN group (7.25 ± 0.89 pg/mL) and NA group (5.54 ± 0.89 pg/mL) compared with sham (1.75 ± 0.89 pg/mL).

KC/GRO (CXCL1) was strongly induced by CBN exposure in plasma (39.21 ± 4.93 pg/mL) and elevated in NA animals (23.98 ± 4.93 pg/mL), both significantly higher than sham (7.06 ± 4.93 pg/mL;). TNF-α levels also increased in NA (4.76 ± 0.31 pg/mL) and CBN (3.64 ± 0.31 pg/mL) groups compared with sham (1.20 ± 0.31 pg/mL), with NA levels significantly higher than CBN, suggesting a more pronounced acute systemic response in the NA group. IL-1β, IL-5, and IL-13 were not detectable (NT) in plasma from sham animals but were measurable in NA and CBN groups; however, differences among groups did not reach statistical significance. The typical detection range for these cytokines in the MSD V-PLEX Rat Pro-inflammatory Panel is 0.61–2500 pg/mL. IFN-γ levels showed no significant differences across groups. Collectively, these findings demonstrate a robust systemic inflammatory response to CBN exposures, with patterns of cytokine elevation distinct from those observed in lung tissue.

In brain tissue, several pro-inflammatory cytokines were significantly altered following NA and CBN exposures, indicating a neuroimmune response to inhaled toxicants (Figure 3). IL-1β levels showed no statistically significant differences among groups. However, IL-6 concentrations were significantly reduced in both NA (39.06 ± 4.15 pg/mL) and CBN (35.54 ± 4.15 pg/mL) groups compared with sham (67.05 ± 4.15 pg/mL), suggesting suppression of this cytokine in the brain. Similarly, IL-10 levels were markedly decreased in NA (5.01 ± 0.90 pg/mL) and CBN (5.89 ± 0.90 pg/mL) groups compared with sham (10.06 ± 0.90 pg/mL).

IL-4 levels in the brain were also reduced by NA (0.43 ± 0.08 pg/mL) and CBN (0.52 ± 0.08 pg/mL) relative to sham (0.68 ± 0.08 pg/mL). IL-5 concentrations decreased significantly in NA (11.29 ± 1.20 pg/mL) and CBN (12.26 ± 1.20 pg/mL) groups compared with sham (18.69 ± 1.20 pg/mL). IL-13 levels were significantly lower in NA (0.78 ± 0.24 pg/mL) and CBN (0.37 ± 0.24 pg/mL) compared with sham (1.40 ± 0.24 pg/mL). Conversely, IFN-γ was significantly elevated in NA (6.29 ± 0.59 pg/mL) and CBN (6.41 ± 0.59 pg/mL) relative to sham (4.73 ± 0.59 pg/mL), indicating immune activation. KC/GRO (CXCL1) was significantly increased in both NA (2.23 ± 0.20 pg/mL) and CBN (2.14 ± 0.20 pg/mL) compared with sham (1.26 ± 0.20 pg/mL), reflecting enhanced chemokine signaling. TNF-α levels were markedly elevated in the NA group (0.68 ± 0.09 pg/mL) and were even higher in the CBN group (1.27 ± 0.09 pg/mL), both significantly increased compared with sham (0.27 ± 0.09 pg/mL), with CBN also significantly higher than NA.

Together, these results reveal that while NA and CBN exposures suppress several key anti-inflammatory and regulatory cytokines in the brain, they simultaneously enhance pro-inflammatory markers like IFN-γ, CXCL1, and TNF-α. These novel findings suggest a shift toward a neuroinflammatory phenotype. An overview of biomarker response for all tissues is summarized in Table 1.

### 2.3. Sample Preparation and Next-Generation Sequencing

RNA-seq analysis was conducted using the CBN and sham exposure groups. While NA exposure induced moderate changes in select cytokines, CBN exposure resulted in significant upregulation of a broader array of pro-inflammatory mediators across lung, plasma, and brain compartments. As the goal of transcriptomic profiling was to identify gene expression pathways strongly associated with response to surrogate burn pit exposure, the CBN group was selected to maximize biological signal and relevance to complex toxicant exposures. The sham group served as a reference for baseline expression. This approach enabled a focused and biologically meaningful comparison between high-response and unexposed animals.

RNA-seq analyses of CBN and sham-exposed rat lung and brain tissues were used to gain unbiased insights into gene expression associated with exposure to CBN. The results identified 278 differentially expressed genes (DEGs) in lungs with FDR-adjusted *p*-values below 0.05 (Appendix A, Column F). In the brain, 19 DEGs were identified with FDR-adjusted *p*-values below 0.05 (Appendix A, Column F). Multidimensional scaling plots visualizing tissue expression dissimilarity between the treatment groups based on the top 200 genes (determined by log2 fold-change) demonstrate the broad effects of CBN exposure compared with sham for lung (Figure 4A) and brain (Figure 4B). The top ten DEGs are labeled in volcano plots of lung (Figure 4C) and brain (Figure 4D), with significant DEGs shown in red (up-regulated with CBN exposure) or blue (down-regulated with CBN exposure). Heat maps demonstrate the 15 most significant DEGs, by FDR, in the lung and brain, respectively, (Figure 4E,F). Biological relevance and function of the top 15 DEGs are summarized in Table 2.

### 2.4. Gene Set Enrichment Analysis

Quantitative changes obtained from RNA-seq were analyzed by Gene Set Enrichment Analysis (GSEA) using gene sets from the Molecular Signatures Hallmark, Kyoto Encyclopedia of Genes and Genomes (KEGG), and the Saul SenMayo gene set for senescence-associated genes to identify molecular pathways that were significantly altered with CBN exposure [129,130,131,132]. The Saul SenMayo gene set represents a validated transcriptional signature of cellular senescence derived from multi-tissue expression data and curated to distinguish senescent from non-senescent cells. Its inclusion enabled detection of senescence-related transcriptional programs not captured by conventional pathway databases, making it particularly useful for toxicant models where cellular stress and aging mechanisms may be engaged [132].

In the lung, CBN exposure significantly activated a variety of pathways including myelocytomatosis proto-oncogene (Myc) version (v) 1 targets, early region 2 binding factor (E2F) transcription factor targets, cell cycle-related and chemokine signaling (Figure 5A). CBN exposure suppressed specific processes in the lung, such as oxidative phosphorylation, Kirsten rat sarcoma viral oncogene homolog (KRAS) signaling, and EMT (Figure 5A). In the brain, GSEA identified enhanced activity in processes involved with EMT, Wingless/Integrated (Wnt) β-catenin signaling, transforming growth factor beta (TGF-β) signaling, and several inflammatory responses such as cytokine-cytokine receptor interaction signaling (Figure 5B). Processes suppressed in brain with CBN exposure include ribosome, spliceosome, and Myc targets v1.

A significant positive association was determined from the brain RNA-seq data using the Saul SenMayo gene set (NES 1.47, q < 0.03), suggesting that gene expression associated with senescence was elevated due to CBN exposures in the brain (Figure 6). Among the senescence-associated genes upregulated in brain tissue following CBN exposure, three were highlighted for their roles in neuroinflammation, neurodegeneration, and blood–brain barrier (BBB) dysfunction. Interleukin 17 (IL-17), a pro-inflammatory cytokine, is known to contribute to BBB disruption, sustained neuroinflammation, and promoting amyloid pathology [133]. AXL receptor tyrosine kinase (Axl) regulates microglial activation and is implicated in neurodegenerative signaling [134]. Endothelial cell-specific molecule 1 (Esm1), a mediator of vascular permeability, suggests BBB compromise [135]. These findings indicate that CBN exposure engages in senescence-related transcriptional programs associated with brain vulnerability.

The Saul SenMayo gene set was not significantly enriched in the lung, but several other molecular pathways identified by GSEA showed consistent alterations in both lung and brain following CBN exposure. Gene sets significantly activated in both tissues included lysosome [KEGG: ko04142], complement [Hallmark], and endocytosis [KEGG: ko04144] (Figure 5) [136,137,138]. These pathways reflect shared biological responses: lysosome activation suggests enhanced clearance of cellular debris and toxic material; complement signaling indicates immune system engagement; and endocytosis reflects increased vesicular trafficking, likely in response to environmental insult. In contrast, oxidative phosphorylation [Hallmark, KEGG: ko00190] was suppressed in both tissues, consistent with impaired mitochondrial energy production and bioenergetic stress [139,140]. Together, these findings suggest a coordinated response to CBN exposure that includes immune activation, cellular adaptation, and mitochondrial dysfunction across lung and brain tissues. In the lung, biological process GO terms significantly enriched among DEGs (FDR *p* ≤ 0.05) further supported immune involvement, including cellular response to tumor necrosis factor, neutrophil and eosinophil chemotaxis, and chemokine-mediated signaling (Appendix A). No significant biological process GO terms were associated with DEGs in the brain.

### 2.5. NF-κB Activation in Lung and Brain Following CBN Inhalation Exposure

NF-κB is a key transcription factor that regulates innate immune responses, inflammation, and cellular stress pathways. Inhaled environmental toxicants such as particulate matter and VOCs found in burn pit emissions activate NF-κB signaling, leading to nuclear translocation of the p65 subunit and transcription of downstream pro-inflammatory genes, including *TNF-α, IL-1β,* and *KC/GRO* (*CXCL1)*. Given its upstream regulatory role, NF-κB serves as a mechanistically informative marker of toxicant-induced immune activation.

To assess NF-κB activation in response to inhalation exposure, we performed western blot analysis of nuclear extracts from lung and brain tissues. CBN exposure significantly increased nuclear NF-κB p65 levels in both tissues compared with sham controls (Figure 7). In the lung, NF-κB p65 abundance was significantly elevated in CBN-exposed animals (Figure 7A,B). This effect was even more pronounced in the brain, (Figure 7C,D).

These results align with the cytokine and RNA-seq analyses showing elevated inflammatory responses across tissues. Notably, GSEA revealed significant alteration of cytokine-mediated responses. Together, these findings confirm that NF-κB activation is a central feature of the lung-brain inflammatory axis following CBN exposure, supporting its utility as a biomarker for inhalation-related immune disruption and dysregulation.

### 2.6. Predictive Modeling Using Elastic Net

To identify a gene signature predictive of toxicant exposure, an elastic net regression model was implemented using the R computing environment [141]. Elastic net was selected for its ability to handle high-dimensional, correlated transcriptomic data by combining L1 and L2 regularization, allowing for sparse but stable gene selection while minimizing overfitting. Compared with lasso, which produced overly sparse models, and ridge regression, which retains all genes, elastic net offers a balanced solution suited to small-sample, high-feature datasets. Using the 278 differentially expressed genes (FDR < 0.05) in the lung, the model selected a 100-gene subset predictive of exposure status. Leave-one-out cross-validation (LOO-CV) demonstrated 100% accuracy in distinguishing CBN-exposed from sham lung samples (Figure 8A,B), indicating the model’s strong classification performance and supporting its utility for molecular biomarker discovery.

Among the selected genes, several were particularly informative in characterizing the lung’s transcriptional response to CBN exposure. We highlight ten genes (IgV, Kcnq3, Sirpb3, Tgfbr1, Tm4sf19, St3gal2, Tiam1, Ccl9, Foxn4, and Tm4sf1) that emerged as the top predictors of exposure status (Figure 8B; Table 3). These genes represent key biological functions involved in immune regulation (IgV, Sirpb3, Ccl9, Tgfbr1, St3gal2), ion channel activity and neurogenic inflammation (Kcnq3), fibrosis and epithelial remodeling (Tgfbr1, Tm4sf19, Tiam1), and tissue repair or regenerative capacity (Foxn4, Tm4sf1). Each gene’s biological function and its potential impact on long-term lung health is summarized in Table 3, which highlights their roles in inflammation, oxidative stress, epithelial damage, and impaired regeneration hallmarks of lung injury [20,27,142]. The diversity of pathways represented in this multigene profile underscores the complex molecular landscape triggered by inhaled toxicants like CBN and reflects converging mechanisms that may drive persistent airway pathology.

In contrast to lung tissue, elastic net modeling using brain RNA-seq data did not add anything to our conclusions. Only 19 genes in the brain met FDR significance thresholds (as compared with 278 in the lung). Using an elastic net model with these 19 genes also had 100% classification accuracy, with the same classification threshold that was used for the lung tissue. However, the elastic net model gave all 19 genes roughly equal weight in the model, and therefore did not help to identify predictive genes over and above what the DEG analysis provided.

## 3. Discussion

These findings expand upon our previous study where we demonstrated that whole-body inhalation of nano-sized CB particles induced systemic inflammatory responses across multiple tissues, including lung, plasma, artery, and brain [16]. Trembley et al. used a single-component particulate exposure and focused on short-term protein-level changes [16]. The current study builds on that foundation by employing a more complex and longer co-exposure to carbon black and naphthalene (CBN) to simulate the multifactorial composition of military burn pit emissions. Importantly, we incorporated transcriptomic profiling in both lung and brain tissues to inform broader and more mechanistically detailed molecular responses.

The data reveal tissue-specific inflammatory patterns across lung, plasma, and brain following CBN exposures. In the lung and plasma, exposure induced robust increases in pro-inflammatory cytokines including IL-4, IL-10, KC/GRO (CXCL1), and TNF-α. Notably, IL-6 was significantly elevated in plasma but suppressed in lung tissue. In contrast, the brain exhibited a divergent cytokine profile characterized by reductions in IL-4, IL-5, IL-6, IL-10, and IL-13, while IFN-γ and KC/GRO were significantly elevated. TNF-α was also markedly increased in the brain, especially with CBN exposure. These distinct inflammatory profiles across tissues underscore the presence of a compartmentalized yet interconnected response, supporting the hypothesis of a functional lung–brain axis in the context of complex inhalational exposures. Increased NF-κB nuclear translocation (~2-fold in lung and >3-fold in brain) supports its central role in mediating systemic inflammatory signaling.

Lung transcriptomic analysis demonstrated broad upregulation of genes involved in immune activation, oxidative stress, and epithelial remodeling. Notable genes included Sirpb3, Tgfbr1, and Tm4sf19 (fibrosis and immune regulation), Kcnq3 (sensory modulation), St3gal2 and Tiam1 (cell migration and glycosylation), and Ccl9, Foxn4, Tm4sf1, and IgV gene segments (adaptive immunity and epithelial repair). Elastic net modeling identified a subset of genes as highly predictive of CBN exposure, forming a distinct lung-specific molecular signature of injury. Gene set enrichment analysis (GSEA) corroborated these findings, with enrichment in chemokine signaling, E2F targets, and suppressed oxidative phosphorylation, indicating active immune responses and mitochondrial stress.

In the brain, transcriptomic changes included enrichment of senescence-associated gene sets (Saul SenMayo), Wnt/β-catenin and TGF-β signaling, and suppression of oxidative phosphorylation. Upregulated genes such as Hist1h4m, Zbtb20, Cyp7b1, and FAM107A were associated with epigenetic modulation, neuroinflammation, and lipid metabolism, while downregulated genes like Chchd4 and Pom121L2 implicated impaired mitochondrial and nucleocytoplasmic function. The enrichment of senescence-associated transcripts exclusively in the brain suggests a lower threshold for senescence induction or reduced regenerative capacity compared with the lung. This may contribute to long-term neurological consequences following toxicant exposure.

These findings align with and extend multiple prior models of DRRD. Hathaway et al. demonstrated mitochondrial collapse and oxidative injury in response to carbon black and ozone [142]. Vance et al. showed phase-dependent respiratory suppression from simulated burn pit emissions but did not investigate transcriptional changes [164]. Our model expands these studies by incorporating both volatile chemical and particulate exposure components thereby revealing molecular evidence for injury in both lung and brain tissues.

Our data also resonate with Gutor et al. (2022), which reported immune-driven airway remodeling in deployment-associated constrictive bronchiolitis, and Gutor et al. (2024), which described a sulfur dioxide (SO_2_) exposure model of post-deployment respiratory syndrome (PDRS) [20,165,166]. The latter study revealed pulmonary vascular remodeling and oxidative injury mediated through the ROS–isoLG–SIRT3–SOD2 axis [166]. While Gutor et al. emphasized vascular pathology and pulmonary hypertension, our CBN model demonstrates overlapping pathways, including Noxo1, Tgfbr1, and NF-κB activation, but uniquely captures CNS involvement, highlighting the added value of a multi-tissue toxicology framework. The combined chemical and PM insults of CBN may induce broader systemic effects than SO_2_ alone. These models are synthesized in the review by Wang et al. (2023), which calls for harmonized inhalation platforms that more accurately reflect deployment-related exposures [9]. Our study contributes to this effort by integrating chemical and PM toxicants complexity, advanced transcriptomic modeling, and neuroinflammatory endpoints.

Epidemiologic studies have consistently linked burn pit exposures to increased risks of psychiatric conditions, cognitive impairment, and neurodegenerative disease [4,5,8,167,168]. Our observation of elevated TNF-α in both the plasma and brain, alongside senescence-linked transcriptional changes, supports the hypothesis that the lung-derived inflammation can disrupt the blood–brain barrier and trigger neuroinflammatory cascades and injury. These immune signals may activate microglia and alter neuronal homeostasis, contributing to the burden of neuropsychiatric disease in exposed Veterans [4,5,8,167,168].

This study has defined biologically coherent alterations in molecular pathways in lung and brain tissues, providing strong evidence for a systemic inflammatory axis following inhalation of burn pit–like emissions. By integrating elastic net modeling, transcriptomic profiling, and cytokine measurement across multiple tissues, our model offers a mechanistic framework for early detection and investigation into DRRDs and neurological diseases following military burn pit exposures. These findings support the expansion of toxicological assessments to include CNS outcomes and provide a foundation for developing clinical diagnostics and therapeutics that address both pulmonary and neuropsychiatric risks in exposed populations.

### Limitations

This study provides valuable insights into the early inflammatory and transcriptional changes induced by surrogate military burn pit exposures. However, several limitations should be acknowledged. First, the exposure protocol used a short-term, subacute model consisting of three 6 h inhalation sessions over three days. While this design simulates an acute deployment-related exposure, it does not capture the chronic, cumulative nature of military burn pit exposure as experienced over many months or years. Longitudinal studies are needed to determine whether these acute molecular signatures persist or evolve with repeated and chronic exposures over time.

Second, although our exposure model incorporates two major toxic components found in burn pit emissions, carbon black (as a particulate matter surrogate) and naphthalene (representing VOCs and PAHs), real-world burn pit smoke consists of a far more complex mixture of chemicals, including dioxins, furans, and heavy metals. As such, our CBN mixture does not fully recapitulate the chemical diversity and additive toxicity of actual burn pit emissions. Further, this study was informed by our earlier work using CB alone; however, the time points used in this study differ from those in our prior work, which limits the ability to make direct comparisons. A significant translational challenge remains the inherent complexity and variability of burn pit toxicant mixtures and exposure durations, which may never be fully replicated experimentally. Expanding preclinical models to include diverse exposure mixtures and administration routes (whole-body inhalation, intranasal, oral) will enhance translational relevance and capture real-world exposure variability.

Third, only male rats were used in this study to reduce variability; however, sex-specific differences in immune and neuroinflammatory responses are well documented [169]. Inclusion of both sexes in future experiments will be essential for understanding sex-based vulnerability to burn pit-associated diseases.

Fourth, the transcriptomic profiling via RNA-seq was limited to a comparison between CBN-exposed and sham animals, excluding the naphthalene-only (NA) group. While this was a data-driven decision based on cytokine profiling (which showed that CBN induced a more robust and distinct inflammatory signature), it limits the ability to distinguish additive or synergistic effects of particulate and volatile toxicants.

Lastly, while the elastic net modeling demonstrated strong discriminatory power in identifying CBN exposure-related gene signatures, external validation in independent cohorts including human data is necessary before these biomarkers can be translated into diagnostic tools. Similarly, additional studies examining the functional consequences and changes in pulmonary physiology in response to differentially expressed genes will strengthen the biological interpretation and generalizability of the predictive signatures.

## 4. Materials and Methods

### 4.1. Experimental Animals

Male Sprague Dawley rats were obtained from Hilltop Laboratories (Scottdale, PA) and housed in an AAALAC-approved facility at West Virginia University (WVU).

### 4.2. Ethics

All procedures were approved by the WVU Institutional Animal Care and Use Committee (protocol 1602000621) and conformed to the most current National Institutes of Health (NIH) Guidelines for the Care and Use of Laboratory Animals. Housing conditions included 12:12 h light/dark cycle, 20–26 °C, 30–70% relative humidity, acclimatization for 48–72 h prior to any procedure, and ad libitum access to food and water.

### 4.3. Whole Body Inhalation Exposure

Rats (sham-control group: age 53 ± 1 days; mass 288 ± 3 g; CBN group: age 53 ± 1 days; mass 303 ± 4 g) were exposed to either HEPA filtered air or a mixture of carbon black (target concentration = 6 mg/m^3^) and naphthalene vapor (target concentration = 10 PPM) for 6 h/day for 3 consecutive days (10 per group). A carbon black-only (CB) group was not included, as the immunological and inflammatory effects of CB alone have been previously characterized in this model using a single 6 h exposure protocol [16]. In contrast, the present study utilized three consecutive days of exposure to better simulate subacute, deployment-relevant inhalation scenarios. The overall objective was to model and characterize the biological effects of a complex mixture of toxicants, reflecting the real-world composition of military burn pit emissions, which consist of both particulate and volatile organic compounds.

Naphthalene alone was included as a comparator group for several reasons: (1) to represent the volatile organic compound (VOC) fraction of burn pit emissions; (2) because rats have been reported to show variable responses to naphthalene, and it was uncertain whether this exposure would elicit an immune response or induce stress in lung, brain, or plasma tissues; and (3) to distinguish effects attributable specifically to naphthalene from those produced by combined particulate and VOC exposure. The naphthalene protocol was selected based on established inhalation conditions previously shown to induce reproducible, though generally less severe, responses in rats [25].

Rats were randomly assigned to either control or experimental groups and were individually housed in cages within a stainless-steel chamber during exposure. The exposure chamber measured 22 × 20 × 20 inches (width × depth × height) with an approximate volume of 144 L. The total airflow through the chamber was approximately 28 LPM during exposures. Bedding material in the bottom of the chamber was soaked with 50 mL of water to provide humidity during exposures. Humidity (30–70%) and temperature (20–26 °C) levels were maintained throughout each exposure period to provide animal comfort.

### 4.4. Preparation of Naphthalene Vapor and Carbon Black Aerosols

Bulk CB powder (Printex 90^®^, composed of 99.9% carbon) was obtained as a gift from Evonik (Frankfurt, Germany) and naphthalene crystals (99% purity, Sigma-Aldrich, St. Louise, MO, USA) were acquired. Before each toxicant exposure, thirty grams of naphthalene were placed into a glass beaker with input and output flow ports. The beaker was placed onto a hot plate set to 60 °C and heated for 30 min prior to exposure to begin sublimation of the naphthalene into a vapor. After the exposure began, a mass flow controller (MFC) regulated the airflow through the beaker and pushed the naphthalene gas through the system. Fifty grams of CB powder was placed into a high-pressure acoustical generator (HPAG, IEStechno, Morgantown, WV, USA) which aerosolized and de-agglomerated the CB powder. A mass flow controller pushed the aerosolized CB out of the generator and into a Venturi pump (JS-60 M, Vaccon, Medway, MA, USA) to further de-agglomerate the particles. The CB aerosols were then mixed with the naphthalene vapor before being passed into the inhalation exposure chamber (Cube 150, IEStechno, Morgantown, WV, USA). Multiple instruments sampled continuously from the chamber to provide real-time, accurate measurements of the aerosol and gas levels. A light scattering device (PDR-1500, Thermo Environmental Instruments Inc., Franklin, MA, USA) was utilized to determine the aerosol mass concentration and a photo ionization detector (MiniRAE 3000, RAE Systems, Sunnyvale, CA, USA) measured the levels of naphthalene vapor within the chamber. Stable concentration levels were maintained in real-time via software feedback loops. The aerosol mass concentration was adjusted by varying the sound energy in the acoustical generator and the naphthalene vapor level was regulated by varying the airflow through the glass beaker. 37 mm PTFE filters were used for gravimetric measures concurrent with the PDR-1500 measures to obtain a calibration factor. These gravimetric measures were also considered the “gold standard” for the mass concentration measurements, and these levels are reported in the Results section. The MiniRAE 3000 was calibrated daily with isobutylene and a correction factor was applied to obtain the correct naphthalene levels. The aerosol and vapor leaving the exposure chamber was HEPA and charcoal filtered before entering the house exhaust. The particle count size distribution of the aerosol was measured using a high-resolution electrical low-pressure impactor (ELPI, Dekati, Tampere, Finland).

### 4.5. Tissue Collection and Processing

Animal tissue harvesting was performed 24 h after the final exposure. Euthanasia of the rats was performed via exsanguination under deep anesthesia (5% induction, 2% maintenance with isoflurane gas) followed by organ removal. Plasma aliquots and whole organ tissues (brain and lung) were snap frozen in liquid nitrogen and stored at −80 °C. Solid tissues were pulverized on dry ice to granular homogenous powder. For each assay (inflammation panel, western blot, RNA-seq), ~100 mg of the same whole organ homogenized tissue from each animal was used to prepare lysate or extract RNA, ensuring that all assays were performed on matched aliquots from the same individual animal.

### 4.6. Inflammation Panel

Plasma samples were diluted in assay buffer, and pulverized brain and lung tissues were homogenized in RIPA buffer (Thermo Fisher Scientific, Waltham, MA, USA). Total protein concentrations in whole tissue lysates were determined using a Direct Detect spectrophotometer (MilliporeSigma, Burlington, MA, USA). Pro-inflammatory cytokines were quantified using the Meso Scale Discovery (MSD) V-PLEX Rat Pro-inflammatory Panel 2 Kit (K15059G; MSD, Rockville, MD, USA), which utilizes a multiplexed immunoassay based on electrochemiluminescence (ECL) detection technology. Concentrations of IFN-γ, IL-1β, IL-4, IL-5, IL-6, IL-10, IL-13, KC/GRO, and TNF-α were measured in brain and lung tissue lysates (50 µg total protein) and diluted plasma samples (25 µL). For all groups, the final above detectable threshold signal sample size ranged from 6 to 10 per group. Data were analyzed using MSD Discovery Workbench 4.0 software [16].

### 4.7. Western Blot Analysis

Western blotting was used to detect and quantify nuclear fractions isolated from homogenously pulverized whole tissue using a nuclear extraction reagent (NE-PER; Thermo Scientific, Waltham, MA, USA). Protein concentrations were determined using an infrared spectrometer (Direct Detect; Millipore, Burlington, MA, USA). 25 µg of protein was loaded onto 4–20% precast Tris-Glycine gels (Bio-Rad, Hercules, CA, USA). The protein was transferred using the Trans-Blot Turbo system (Bio-Rad, Hercules, CA, USA). Total protein was stained using LI-COR Revert 700 stain (LI-COR, Lincoln, NE, USA) and imaged using the LI-COR Odyssey FC imager (LI-COR, Lincoln, NE, USA). Blocking was performed using Intercept Blocking Buffer (LI-COR, Lincoln, NE, USA). NF-κB primary antibody was used at 1:1000 dilution (Cell Signaling Technology 6959; Danvers, MA, USA) with incubation performed overnight at 4 °C with agitation. Secondary antibody was used at 1:20,000 dilution for a concentration of 1.0 × 10^−1^ µg/mL (goat anti-mouse LI-COR 926-32210; LI-COR, Lincoln, NE, USA). Densitometry was performed using Image Studio software (LI-COR, Lincoln, NE, USA).

### 4.8. Data Analysis

GraphPad Prism 10.2.3 was used for the biomarker and immunoblot data calculations and bar plot visualizations (GraphPad, San Diego, CA, USA). Descriptive statistics are provided for CBN aerosol characterization. MSD data were analyzed using ordinary one-way ANOVA followed by Holm-Šídák’s multiple comparisons test (5–10 per group). The differences between sham-control and CBN-exposed groups for western blot data were identified using unpaired *t*-test or Mann–Whitney tests according to data distribution (8 per group). Significance was established as *p*-value ≤ 0.05 providing strong evidence against the null hypothesis. 

### 4.9. RNA Sequencing

Tissue and resulting data were processed using a custom pipeline and normalized as previously described [170]. Briefly, the RNA was isolated from homogenously pulverized whole lung and brain tissues by the University of Minnesota Genomics Center using a commercial kit (RNeasy Plus Universal kit, Qiagen, Germantown, MD, USA) according to manufacturer’s instructions. RNA was quantified using a RiboGreen assay (Thermo Fisher, Waltham, MA, USA), and the quality was assessed by capillary electrophoresis using a Tapestation (Agilent, Santa Clara, CA, USA). All samples had an RNA integrity number (RIN) ≥ 8. Libraries were prepared using an Illumina TruSeq Stranded Total RNA Library Prep Kit (San Diego, CA, USA), and sequencing was performed using a Novaseq 6000 (Illumina) to generate 150 bp read pairs. Read quality control, trimming, and alignment were performed separately for lung and brain tissues using a custom analysis pipeline (Pipeline for UMII/RI RNASeq Analysis), developed and maintained by the Research Informatics (RI) group at the Minnesota Supercomputing Institute (MSI). Adaptor sequences were trimmed from raw reads using Trimmomatic (v0.33), and the quality of raw and trimmed reads was assessed with FastQC (v0.11.7). Ribosomal RNA (rRNA) contamination, which was low in all samples (≤1%), was quantified by aligning a subsample of 10,000 reads pairs from each sample to rRNA sequences from the SILVA database (release 132). Trimmed reads were aligned to the Rattus norvegicus reference genome (mRatBN7.2, Ensembl release 109) using HISAT2 (v2.1.0). Reads with mapping quality < 60 were removed. Raw counts were generated using the featureCounts tool from the Subread package (v1.6.2).

### 4.10. Differential Gene Expression Analysis

For each tissue, raw counts were imported into R (v4.3.0). Genes shorter than 200 bp or with very low expression (<1 CPM in at least 50% of samples) were pruned from the count matrix before differential expression testing. Pairwise differential expression testing between CBN-exposed and sham groups was performed separately for brain and lung tissues using a likelihood ratio (LR) test or quasi-likelihood F (QLF)-test, respectively, using EdgeR (v4.0.5). Genes with a false discovery rate (FDR)-adjusted *p*-value ≤ 0.05 were considered significantly differentially expressed genes (DEGs, Appendix A).

### 4.11. Gene Set Enrichment Analysis

Expressed genes from brain or lung tissues were ranked by multiplying −log_10_ (*p*-value) by the sign of the fold-change from differential expression testing, where higher-ranked genes have higher expression in CBN-exposed than sham tissues and lower-ranked genes have lower expression in CBN-exposed than sham tissues. Gene set enrichment analysis (GSEA) using the GSEA function from the clusterProfiler R package (v4.10.0) was performed using the pre-ranked genes and combined gene sets that included rat Hallmark gene sets (MSigDB), the Saul SenMayo gene set with human gene symbols mapped to rat symbols (MSigDB), and KEGG pathways. Hallmark and KEGG pathways were not included in the GSEA if they were not applicable to brain or lung tissues. For example, the Alzheimer disease KEGG pathway was removed for lung, and the Hallmark bile acid metabolism pathway was removed for both tissues. For all GSEA, only gene sets with at least 10 genes and at most 500 genes were analyzed. Overrepresentation analysis with all biological process gene ontology (GO) terms was performed separately for both tissues using the topGO R package (v2.58.0) with the default weight01 algorithm. Gene sets and GO terms with q-value ≤ 0.05 were considered significantly enriched/overrepresented. GSEA summary plots were created using the ggplot2 R package (v 3.5.2), heatmaps were produced using the pheatmap R package (v 1.0.12), and individual pathway enrichment plots were produced using the R package enrichplot (v 1.24.2).

### 4.12. Predictive Modeling Using Elastic Net

To build a predictive model of CBN exposure we extended our analysis to include computational methods to identify predictors that remain associated with exposure after accounting for correlations with the expression of other genes. To achieve this, a penalized regression model, which adds a penalty for models with more predictors, was applied to the lung tissue expression data for 278 differentially expressed genes (FDR *p*-value ≤ 0.05) [171]. Penalized regression helps select a subset of the most predictive genes, in this case genes predictive of CBN exposure, by reducing the influence of less relevant ones [172]. Different methods have been developed to exclude or down-weight the importance of genes to produce a robust, predictive signature that depends on a smaller subset of genes. We tested several methods, including lasso, elastic net, and ridge regression, with lasso being the most conservative and ridge regression the least conservative [173]. Elastic net, which selected 100 genes, was chosen as a balanced option given the small sample size. An extended explanation of this model is included in the Supplemental Methods [172,174,175,176,177].

### 4.13. Availability of Data and Materials

Most data generated or analyzed during this study are included in this published article and its Appendix A files. Raw fastq files and processed data files containing raw counts and trimmed mean of M-values (TMM) normalized counts per million (CPM) and log_2_ CPM values (produced by EdgeR) for all samples are available through the Gene Expression Omnibus (GEO) accession number GSE281993. Other data is available from the corresponding author on reasonable request.

## 5. Conclusions

This study highlights key biological mechanisms linking inhaled toxicants from burn pit exposures to systemic inflammation and multi-organ dysfunction. The use of a short-term, three-day exposure window was intentionally designed to capture early inflammatory and transcriptomic shifts, providing insight into the sub-acute biological triggers that may precede and contribute to longer-term health effects. These findings advance our mechanistic understanding of the initial molecular responses to complex airborne toxicant mixtures, which is critical for early risk assessment and intervention.

By characterizing molecular and immune signatures in both lung and brain tissues, our results support efforts to develop toxicological risk assessments, enable earlier diagnosis, and improve targeted care for Veterans exposed to military burn pit emissions. Importantly, the 100-gene lung signature identified through elastic net modeling demonstrates strong potential for future development as a biomarker panel to objectively assess toxicant exposure and injury, guide disease stratification, and facilitate therapeutic testing. Further validation in independent cohorts and refinement to a clinically practical subset will be essential next steps toward translation of these findings into diagnostic tools.

We also observed selective enrichment of the Saul SenMayo senescence signature in brain, but not lung, following exposure. This finding suggests that the brain may be more sensitive to acute or subacute inhalational toxicant exposures, potentially due to its unique cellular composition, high metabolic activity, and limited regenerative capacity. However, additional factors such as timing of sample collection or limitations of the gene set may also contribute to this tissue specificity. Addressing these uncertainties will require future studies employing time-course analyses that encompass not only subacute but also long-term chronic exposures, as well as single-cell RNA sequencing and spatial transcriptomic approaches, to better resolve tissue-specific and cell-type–specific responses.

These insights align with broader public health goals to mitigate the long-term effects of air pollution and occupational exposures [27,142,178]. Incorporating digital biomarkers, non-invasive imaging technologies, and integrated diagnostics represents a long-term translational goal aimed at enhancing our ability to detect, monitor, and understand burn pit-related health effects. These advanced approaches are not yet widely implemented but hold promise as scalable, non-invasive tools to assess physiological and inflammatory responses in affected populations [2,18,179]. Embedding such innovations within a broader community of care model uniting basic scientists, environmental toxicologists, clinicians, and the Veteran communities we serve ensures that the scientific discovery is rapidly translated into meaningful, accessible healthcare solutions [2,179,180]. This model reinforces the value of interdisciplinary collaboration to address the complexity of environmental toxicant exposures [2,18,179,180].

## Figures and Tables

**Figure 1 ijms-26-07238-f001:**
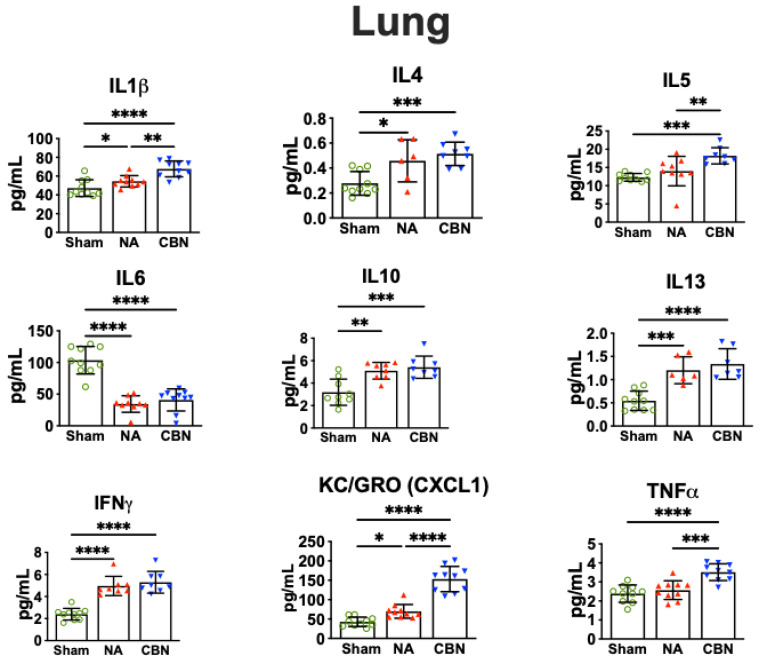
Pro-inflammatory protein profiling in lungs. Lung tissues were collected 24 h post-exposure and analyzed using a multiplex electrochemiluminescence assay (MSD V-PLEX Rat Pro-inflammatory Panel 2). Concentrations of IL-1β, IL-6, IFN-γ, IL-4, IL-10, KC/GRO (CXCL1), IL-5, IL-13, and TNF-α were quantified. Data were analyzed using ordinary one-way ANOVA followed by Holm-Šídák’s multiple comparisons test. Statistical significance is indicated as follows: * *p* < 0.05, ** *p* < 0.01, *** *p* < 0.001, and **** *p* < 0.0001. Error bars represent mean ± SEM, 6–10 samples per group. Treatment group symbols: green circles, sham; red triangles, NA; blue triangles, CBN.

**Figure 2 ijms-26-07238-f002:**
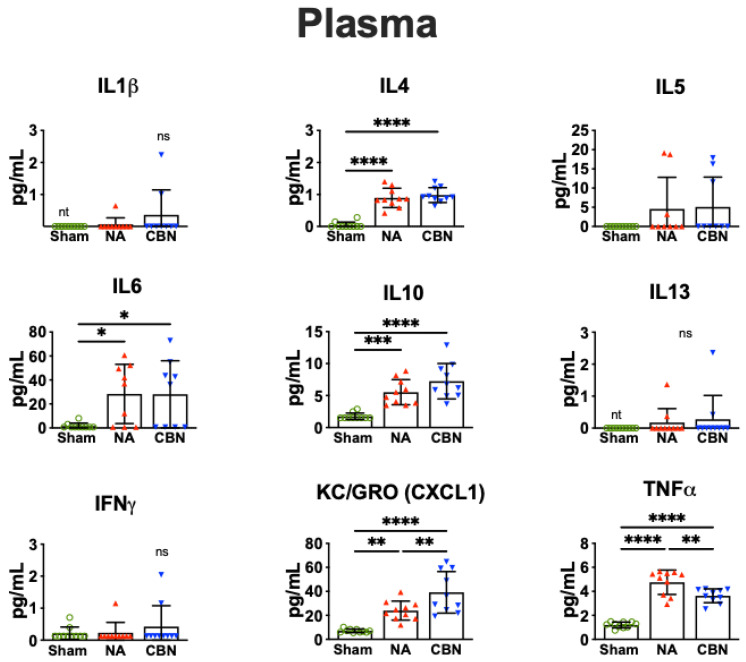
Pro-inflammatory protein profiling in plasma. Plasma was collected 24 h post-exposure and analyzed using a multiplex electrochemiluminescence assay (MSD V-PLEX Rat Pro-inflammatory Panel 2). Statistical significance was determined by ordinary one-way ANOVA with Holm-Šídák’s post hoc test and is denoted as follows: * *p* < 0.05, ** *p* < 0.01, *** *p* < 0.001, and **** *p* < 0.0001. Error bars represent mean ± SEM. nt, not detected. ns, not significant, 6–10 samples per group. Treatment group symbols: green circles, sham; red triangles, NA; blue triangles, CBN.

**Figure 3 ijms-26-07238-f003:**
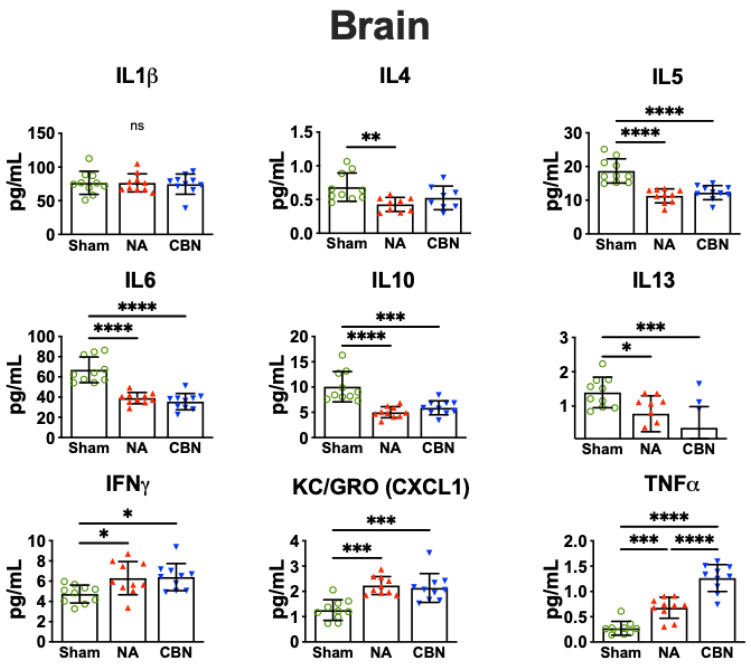
Pro-inflammatory protein profiling in the brain. Brain tissues were harvested 24 h post-exposure and analyzed using a multiplex electrochemiluminescence assay (MSD V-PLEX Rat Pro-inflammatory Panel 2). Statistical significance was determined by ordinary one-way ANOVA with Holm-Šídák’s post hoc test and is denoted as follows: * *p* < 0.05, ** *p* < 0.01, *** *p* < 0.001, and **** *p* < 0.0001. Error bars represent mean ± SEM. ns, not significant, 6–10 samples per group. Treatment group symbols: green circles, sham; red triangles, NA; blue triangles, CBN.

**Figure 4 ijms-26-07238-f004:**
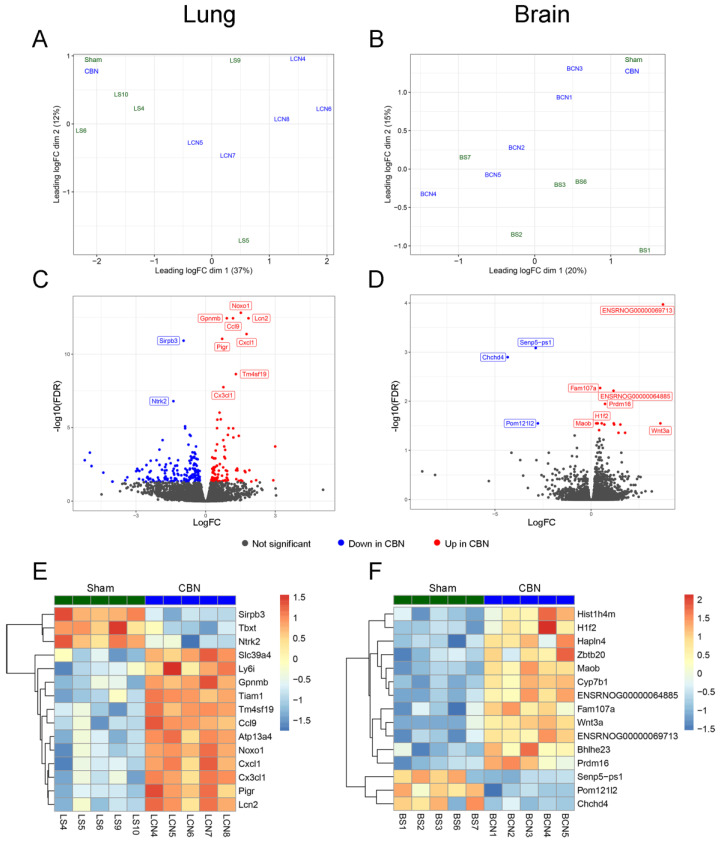
Summary of RNA sequencing differential expression analysis. Multidimensional scaling plots showing differences between rats within each tissue type based on expression of 200 genes with the greatest log2 fold-change difference between groups treated with sham air (control) and CBN for lung (**A**) or brain (**B**). Volcano plots summarizing differential expression results. Significantly differentially expressed (DE) genes (FDR *p*-value < 0.05) are shown in red (up-regulated with CBN exposure) or blue (down-regulated with CBN exposure), and the top ten DE genes are labeled for lung (**C**) or brain (**D**). Heatmap showing scaled log2 counts per million (z-scores) for the top 15 DE genes in lung (**E**) or brain (**F**).

**Figure 5 ijms-26-07238-f005:**
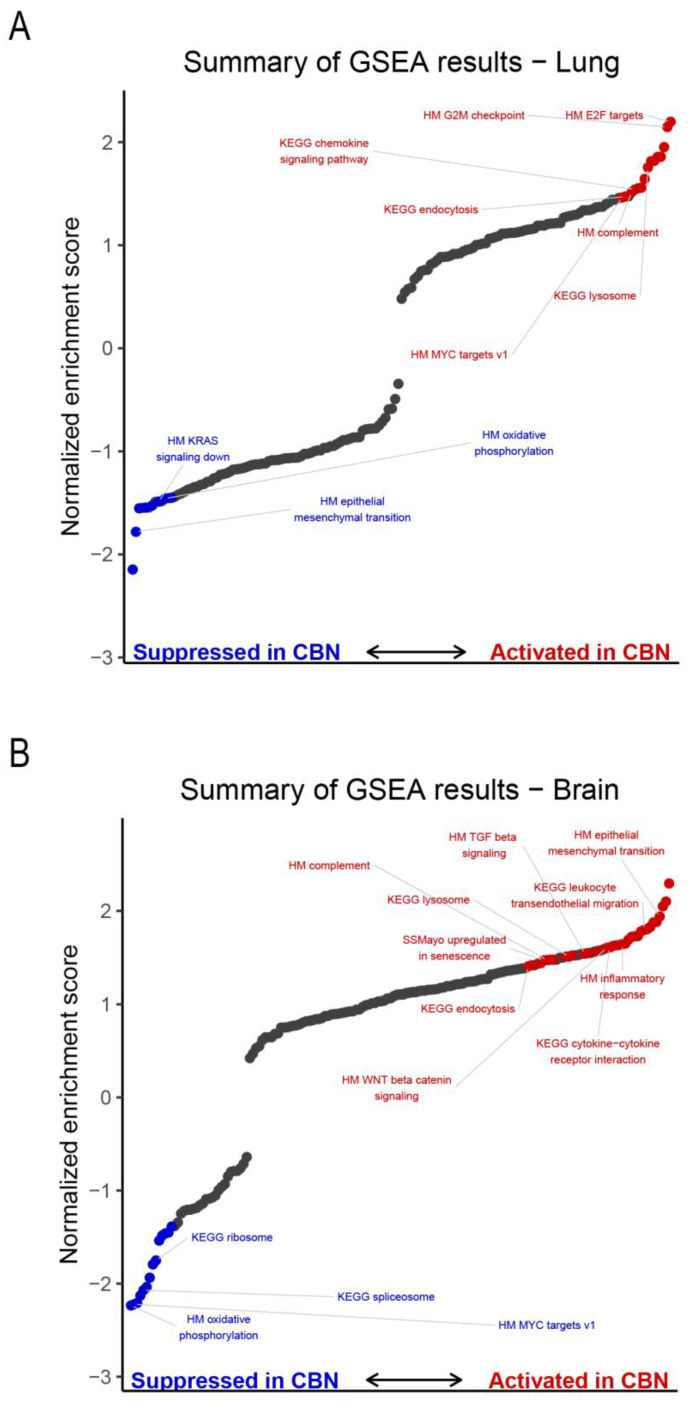
Summary of gene set enrichment analysis (GSEA) results. GSEA was performed separately in brain and lung tissues using Hallmark (HM prefix), KEGG (KEGG prefix), and the Saul Sen Mayo (SSMayo prefix) (genes upregulated in senescent cells) gene sets. Normalized enrichment scores are shown for all gene sets tested in lung (**A**) and brain (**B**) with significantly enriched gene sets colored in red, activated in carbon black naphthalene (CBN)-treated, or blue, suppressed in CBN-treated samples).

**Figure 6 ijms-26-07238-f006:**
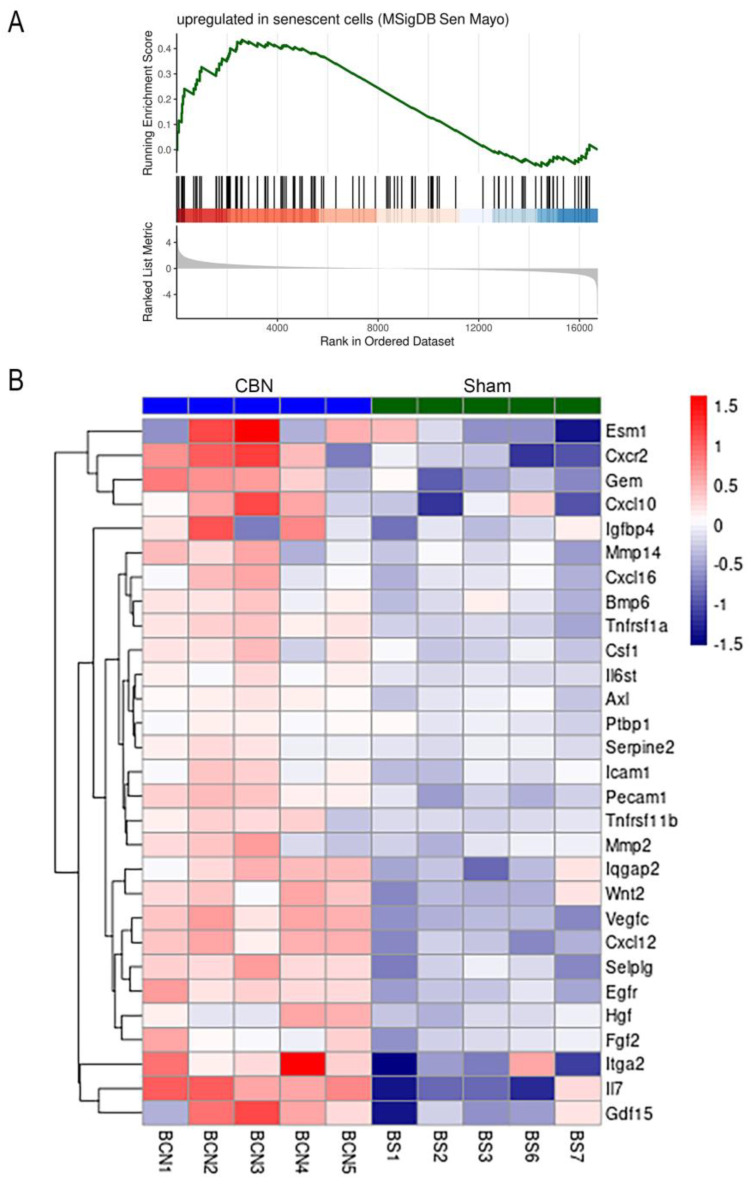
The Saul Sen Mayo gene set (genes upregulated in senescent cells) was significantly activated in brains of CBN-treated rats. (**A**) Enrichment plot showing the running enrichment score for genes in the Saul Sen Mayo gene set using a ranked list of all expressed genes based on results of differential expression analysis between CBN-treated and sham control brain tissue. (**B**) Heatmap showing log_2_ fold-changes (log_2_FC) of each sample over the mean of the comparison group for leading-edge genes (genes that contribute to the enrichment signal) from the Saul Sen Mayo gene set. Genes are ordered in the heatmap by hierarchical clustering.

**Figure 7 ijms-26-07238-f007:**
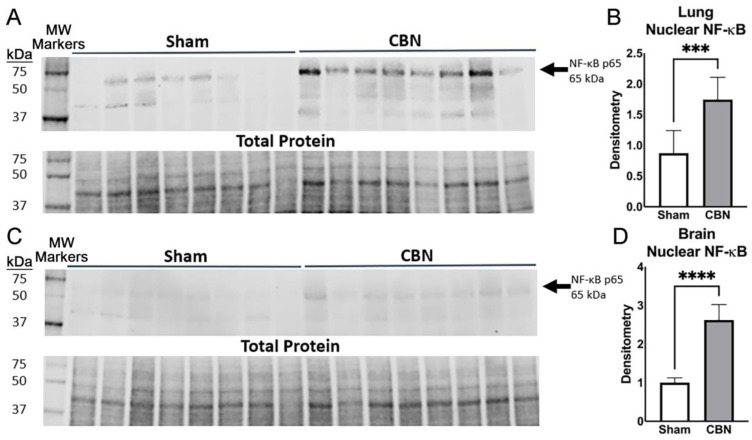
Nuclear NF-κB p65 protein levels are elevated in lung and brain following CBN exposure. Western blot and densitometry analyses of NF-κB p65 in nuclear extracts from rat lung (**A**,**B**) and brain (**C**,**D**) tissues following exposure to CBN or sham control. (**A**) Western blot of NF-κB p65 (65 kDa) in lung nuclear extracts from sham and CBN-exposed rats. Total protein is shown as a loading control. Molecular weight standard (in kDa) shown in the first lane. (**B**) Quantification of NF-κB p65 nuclear abundance in the lung shows a significant increase in the CBN-exposed group compared with sham (*** *p* < 0.001). (**C**) Western blot of NF-κB p65 in brain nuclear extracts with total protein loading control. (**D**) Densitometry analysis reveals a robust increase in brain nuclear NF-κB p65 levels following CBN exposure (**** *p* < 0.0001). Molecular weight (MW) in kilodaltons (kDa) markers are shown on the left side of the blot (**A**,**C**). Densitometry values were normalized to total protein per sample. Data are presented as mean ± SEM.

**Figure 8 ijms-26-07238-f008:**
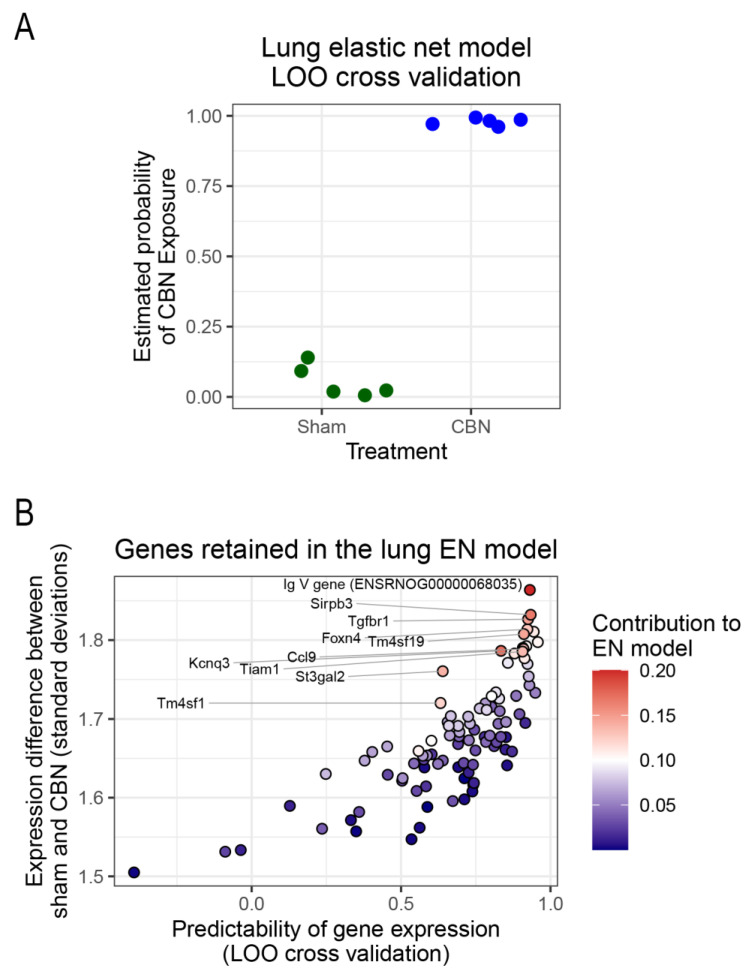
Elastic net regression model predicts CBN exposure based on gene expression profiles in lung. (**A**) Binary classification plot showing the estimated probability that each biological replicate belongs to the CBN exposure group versus the sham control, based on elastic net modeling using 278 differentially expressed genes (DEGs) between CBN and sham control groups. The prediction for each sample was based only on the data from the other samples (leave-one-out cross-validation). The model’s predictive accuracy depends on the threshold for dichotomizing the estimated probability of belonging to the CBN exposure group into “predict CBN”/”predict Sham” outcomes. The elastic net model is 100% accurate when a classification threshold of 0.50 is used. Symbols for treatment groups: green circles, sham; blue circles, CBN. (**B**) This plot illustrates the relationship between gene predictability (x-axis) and standardized expression difference (y-axis) for the final 100 genes retained in the elastic net model. Each point represents a gene, colored by its relative contribution to the model. The top 10 predictive genes are labeled.

**Table 1 ijms-26-07238-t001:** Protein biomarkers altered by inhalation exposure and biological relevance.

Biomarker	Tissue	NA	CBN	Biological Role/Relevance
IL-1β	Lung	↑	↑	IL-1β is a pro-inflammatory cytokine involved in acute lung inflammation with key roles in local and systemic inflammation and is commonly elevated following particulate matter exposure [16,26,27].
IL-4	Lung Plasma Brain	↑ ↑ ↓	↑ ↑ ↓	IL-4 is a Th2 cytokine that plays a role in allergic airway inflammation, pulmonary hypertension and in regulating neuroimmune signaling [16,28,29].
IL-5	Lung Brain	↑ ↓	↑ ↓	IL-5 promotes eosinophil activity; upregulated in lung but suppressed in brain, suggesting compartmentalized immune activation [16,30,31].
IL-6	Lung Brain Plasma	↓ ↓ ↑	↓ ↓ ↑	IL-6 is a pleiotropic cytokine; suppressed in lung and brain, but elevated in plasma during systemic inflammation from inhaled toxicants [32,33,34,35]. IL6 deficiency in acute lung injury is associated with acute respiratory distress syndrome (ARDS) [32,36]
IL-10	Lung Plasma Brain	↑ ↑ ↓	↑ ↑ ↓	IL-10 is an anti-inflammatory cytokine; increased in plasma and lung but decreased in brain, suggesting differential regulation under toxic stress [16,37].
IL-13	Lung Brain	↑ ↓	↑ ↓	IL-13 mediates mucus production and drives fibrosis and remodeling in lung and has context-dependent pro- or anti-inflammatory effects in the brain [28,38,39]. Decreased IL-13 in the CNS is linked to increased neurodegeneration, highlighting how toxicant exposures differentially regulate lung and brain immune responses [39].
IFN-γ	Lung Brain	↑ ↑	↑ ↑	IFN-γ, a Th1 cytokine, plays a dual role in lung and brain toxicity. In lung, it promotes macrophage activation and inflammation, contributing to tissue injury after toxicant exposure and supporting neuroprotective immune regulation [9,16,40]. In brain, IFN-γ both exacerbates neuroinflammation and supports neuroprotective immune regulation [9,16,40].
KC/GRO (CXCL1)	Lung Plasma Brain	↑ ↑ ↑	↑ ↑ ↑	KC/GRO (CXCL1) recruits neutrophils, is expressed by macrophages, and is markedly elevated in lung, plasma, and brain after exposures, a hallmark of lung and brain injury [8,41,42,43,44,45,46].
TNF-α	Lung Plasma Brain	↑	↑	TNF-α is a key pro-inflammatory mediator that drives lung injury through macrophage activation and neutrophil recruitment and promotes microglial activation and neurodegeneration in the brain following toxic exposures [8,9,16,47,48].
NF-κB (western blot)	Lung Brain	N/A N/A	↑ ↑	NF-κB is a transcription factor activated by oxidative and inflammatory stress, driving lung and brain injury [16,49]. PM exposures upregulate NF-κB in lung, promoting inflammation and epithelial damage [49]. In brain, NF-κB activation contributes to neuroinflammation, metabolic dysfunction, and neurodegeneration [47,50].

↑, indicates significantly increased levels; ↓, indicates significantly decreased levels.

**Table 2 ijms-26-07238-t002:** Top 15 DEGs in lung and brain: relevance to toxicant-induced injury.

LUNG		
Gene Symbol Decreased or Increased	Full Gene Name	Biological Function and Disease Relevance
*Sirpb3*	Signal-regulatory protein beta-3	*Sirpb3* is a non-coding pseudogene in the signal regulatory protein (SIRP) family, which includes receptors expressed on myeloid cells. Although it does not encode a protein, *Sirpb3* may influence immune signaling through RNA-based mechanisms, similar to *SIRPB1*, which promotes myeloid activation and inflammation [51]. Persistent activation of SIRP-related pathways can amplify myeloid signaling, driving chronic inflammation, impaired epithelial repair, tissue damage, and fibrosis. In contrast, reduced expression may impair phagocytic clearance and antigen resolution, increasing susceptibility to infection and promoting maladaptive lung remodeling [51].
*Tbxt*	T-box transcription factor T; alias, brachyury	Fundamentally important for lung development, reactivation in adult tissue is associated with lung injury responses, fibrosis via TGF-β, Wnt signaling, and cancer, influencing tissue remodeling and repair outcomes [52,53]. Downregulation of *Tbxt* in the lung may impair epithelial regeneration and repair of injured airways, leading to persistent epithelial damage. It also disrupts necessary matrix remodeling, which can result small airway injury and contribute to chronic inflammation and may increase the risk of diseases like bronchiolitis obliterans and early-stage COPD [54,55].
*Ntrk2*	Neurotrophic receptor tyrosine kinase 2	Encodes for neurotrophic receptor tyrosine kinase 2 (TrkB). Reduced expression of Ntrk2 (TrkB) may impair neurotrophic signaling and epithelial repair, potentially hindering recovery from lung injury. In contrast, increased Ntrk2 expression is associated with aggressive tumor behavior, enhanced metastasis, and poor survival in lung adenocarcinoma [56,57].
*Slc39a4*	Solute carrier family 39 (zinc transporter), member 4; alias, Zrt like protein-4 (*Zip4*)	Encodes for zinc transporter-4 (Zip-4) that is overexpressed in non-small cell lung cancer, where it promotes metastasis by activating epithelial–mesenchymal transition (EMT)-related pathways [58]. ZIP4 promotes NSCLC progression and metastasis by upregulating the snail-N-cadherin signaling axis, thereby facilitating EMT and enhancing cell migration and invasion [59].
*Ly6i*	Lymphocyte antigen 6 complex, locus I	Surface marker selectively expressed in a dysfunctional subpopulation of alveolar type II cells in emphysematous lungs. Ly6i+ cells exhibit impaired regenerative capacity, increased senescence, and inflammation, contributing to progressive tissue damage in COPD [60]. Its presence may lead to long-term respiratory decline and increased risk of chronic lung failure [60].
*Gpnmb*	Glycoprotein non-metastatic melanoma protein B	Transmembrane protein expressed in injured lung epithelial and immune cells, where it regulates cell adhesion, immune response, and tissue repair. In lung injury and cancer, it promotes tumor growth and invasion through membrane signaling and its shed ectodomain, activating integrins and MMPs. Up-regulation of *Gpnmb* in lung leads to its accumulation in fibrotic ECM. Enhances fibroblast proliferation, migration, and fibrotic protein production, contributing to progressive tissue remodeling [61,62].
*Tiam1*	T-cell lymphoma invasion and metastasis 1	Regulates cell migration and cytoskeletal remodeling through Rac1 signaling. Tiam1 may also influence fibroblast activation and endothelial barrier function during lung injury [63,64]. Long-term dysregulation of Tiam1 affects lung health in a disease-dependent manner. Overexpression promotes tumor invasion and metastasis in both small cell and non-small cell lung cancer (NSCLC), while reduced expression may impair tissue repair and endothelial barrier integrity [63,64].
*Tm4sf19*	Transmembrane 4 L six family member 19; alias, transmembrane 4L	Transmembrane protein expressed in lung epithelial cells, where it participates in signal transduction, transcriptional regulation, and response to oxidative stress. It influences GABP-mediated YAP signaling pathways in maintaining epithelial integrity [65]. May also regulate immune cell interactions, contributing to immune surveillance and inflammation in the lung tissue [66]. Altered *Tm4sf19* expression is linked to NSCLC tumor progression and may also contribute to inflammation or epithelial dysfunction in COPD and lung fibrosis. Its effects depend on cellular stress and immune signaling, potentially protecting or worsening disease based on context [65,66].
*Ccl9*	Chemokine (C-C motif) ligand 9; alias, macrophage inflammatory protein-1 gamma (MIP-1γ)	Chemokine expressed in lung tissue that recruits monocytes, macrophages, and dendritic cells via the CCR1 receptor [67] to support innate immune responses during lung injury or infection [68]. Long-term dysregulation of Ccl9 in the lung contributes to chronic inflammation and immune-driven tissue remodeling. Its persistent upregulation is associated with fibrosis and may worsen outcomes in diseases like asthma, COPD, and pulmonary fibrosis. Ccl9 may also amplify lung responses to environmental exposures, promoting progressive damage over time.
*Atp13a4*	ATPase 13A4	P-type ATPase that transports polyamines and cations using ATP hydrolysis; helps maintain ion balance and may support epithelial integrity, stress responses, and signaling [69]. Its regulation of calcium and polyamines suggests a role in oxidative stress and tissue remodeling [70]. *Atp13a4* amplification has been linked to lung cancer prognosis [70]. Dysregulation of polyamine or ion transport via ATP13A4 could also contribute to epithelial dysfunction [70].
*Noxo1*	NADPH oxidase organizer 1	Regulates ROS generation in lungs; elevated in response to air pollution and oxidative injury [71]. Oxidative stress regulator implicated in fibrosis and progressive lung disease such as COPD [72]
*Cxcl1*	C-X-C motif chemokine ligand 1; alias, KC/GRO	Chemokine that recruits neutrophils to inflamed lung tissue; key in early lung immune response to toxins [73,74].
*Cx3cl1*	C-X3-C motif chemokine ligand 1; alias, fractalkine	Chemokine expressed by lung epithelial, endothelial, and smooth muscle cells in both membrane-bound and soluble forms. It regulates monocyte and macrophage recruitment through the receptor Cx3cr1, supporting immune cell adhesion, cytokine release, and tissue remodeling [75,76]. In chronic lung diseases like COPD, asthma, and IPF, dysregulated CX3CL1 expression drives persistent macrophage-mediated inflammation and fibrosis, contributing to progressive tissue damage [75,76]. Elevated protein expression in lung tissue demonstrated in preclinical burn pit models [16].
*Pigr*	Polymeric immunoglobulin receptor	Immunoglobulin receptor (PIGR) transports IgA and IgM across lung epithelial cells to generate secretory IgA at the mucosal surface, where it neutralizes inhaled pathogens and toxins. PIGR expression is regulated by inflammatory cytokines, enabling immune defense and limiting immune response [77,78,79]. Loss of PIGR regulation may increase susceptibility to infection and chronic respiratory illness. Overexpression in the lung often reflects epithelial stress or immune activation, as observed in COPD, asthma, IPF, and lung cancer. While initially protective, sustained PIGR elevation can drive epithelial remodeling, disrupt immune tolerance, and promote chronic inflammation, contributing to airway hyperreactivity and fibrosis [78,79]
*Lcn2*	Lipocalin-2	Iron-binding glycoprotein produced by lung epithelial and immune cells, playing key roles in inflammation, innate immunity, and iron regulation. While it helps protect against bacterial infection by sequestering iron, excessive LCN2 expression contributes to chronic inflammation, oxidative stress, and lung tissue damage [80,81]. Upregulated in both COPD and constrictive bronchiolitis, contributing to inflammation, immune dysregulation, and airway structural damage. Elevation can lead to iron imbalance and oxidative stress, chronic tissue injury and impaired repair. Can promote airway remodeling, fibrosis, and lung function decline, increasing the risk of lung cancer [80,82,83].
**BRAIN**		
**Gene Symbol (Accession)**	**Full Gene Name**	**Biological Function and Disease Relevance**
*Hist1h4m*	Histone cluster 1 H4 family member	Histone protein linked to epigenetic modulation under stress; altered in neurotoxic exposure models [84,85,86]. H4-mediated epigenetic alteration may contribute to memory deficits and aging-related brain dysfunction [84,85,86].
*H1f2*	H1 histone family member 2	Histone protein involved in chromatin structure; dysregulation linked to neurodegenerative changes [84,86,87]. Altered expression found in neuroinflammatory and cognitive impairment models [84,86,87].
*Hapln4*	Hyaluronan and proteoglycan link protein 4; alias, brain link protein 2 (*Bral2*)	Encodes an extracellular matrix protein that links hyaluronic acid to proteoglycans, supporting synaptic architecture and stability [88]. Enriched at GABAergic synapses, *Hapln4* plays critical roles in nervous system development, synaptic maturation, and maintaining extracellular homeostasis [88]. Elevated *Hapln4* expression can impair synaptic transmission, restrict neuroplasticity, and promote neuroinflammation following injury or toxicant exposure, increasing vulnerability to cognitive dysfunction and neurodegenerative diseases [89,90].
*Zbtb20*	Zinc finger and BTB domain containing 20	Transcription factor essential for brain development, regulating neuronal differentiation, synaptic plasticity, and memory formation [91]. It also modulates oxidative stress and inflammatory responses, positively regulating pro-inflammatory cytokines during neuroinflammation and injury [92]. Linked to Alzheimer’s pathology and cognitive decline after toxic exposures [93,94].
Maob	Monoamine oxidase B	Encodes monoamine oxidase B, a mitochondrial enzyme that degrades neurotransmitters like dopamine and serotonin, regulating neurotransmitter balance and producing hydrogen peroxide [95,96,97]. Its activity, enriched in astrocytes and neurons, links it to oxidative stress in the brain [98]. Dysregulated MAOB activity leads to elevated oxidative stress, mitochondrial dysfunction, and neuronal damage [98]. Upregulation of *Maob* has been implicated in neurodegenerative diseases such as PD and AD, contributing to cognitive decline, neuroinflammation, and progressive dopaminergic neuron loss [98].
*Cyp7b1*	Cytochrome P450 family 7 subfamily B member 1; alias, oxysterol 7-alpha-hydroxylase brain	Encodes a cytochrome P450 enzyme that plays a crucial role in the metabolism of neurosteroids and cholesterol-derived molecules within the brain [99]. It helps regulate the synthesis of specific oxysterols and neuroactive steroids, impacting neuronal protection, signaling, and inflammation. Dysregulation of *Cyp7b1* can increase vulnerability to neurodegeneration, inflammation, and lipid imbalances seen in conditions like Alzheimer’s disease [99,100,101,102].
ENSRNOG00000064885 (only Ensembl ID available)	Predicted Ensembl rat gene	Predicted to encode a ribosomal protein that maintains ribosome integrity, facilitates translation, and participates in ribonucleoprotein complexes essential for protein synthesis [103]. Ribosomal dysfunction may impair neuronal protein synthesis and plasticity, increasing susceptibility to oxidative stress, inflammation, and neurodegeneration after toxicant exposure [104].
*Fam107a*	Family with sequence similarity 107 member A; alias, Down-Regulated in Renal Cell Carcinoma 1 (*Drr1)*	Stress-inducible, actin-binding protein that regulates cytoskeletal dynamics, supporting synaptic function, neural cell survival, and adaptation to stress [105]. *Fam107a* also modulates microglial activation, linking neural stress responses to neuroinflammation [106]. Sustained upregulation supports neural protection and reduced inflammation, but chronic dysregulation may cause maladaptive inflammation or synaptic dysfunction. Balanced Fam107a expression is crucial for long-term brain health [106].
*Wnt3a*	Wingless/Integrated (Wnt) family member3A	Protein coding gene for Wnt3A glycoprotein involved in Wnt signaling. Plays a vital role in brain development and function by promoting neurogenesis, supporting neuronal differentiation and synaptic plasticity, protecting against oxidative stress, and maintaining neural stem cell niches [107]. *Wnt3a* acts as a double-edged sword in the adult brain, supporting repair, neurogenesis, and immune regulation under balanced activation, but potentially exacerbating neuroinflammation and degeneration when overactivated, particularly in chronic disease or toxic exposures [108].
ENSRNOG00000069713 (only Ensembl ID available)	Predicted Ensembl rat gene	Predicted to function as a long non-coding RNA (lncRNA), Unknown experimentally, but typical lncRNAs regulate gene expression, chromatin remodeling, RNA stability, and may affect ribosome-related processes indirectly [109]. Dysregulation of lncRNAs may impact gene expression to impair memory, plasticity, and promote neurodegeneration after toxicant exposures [110,111].
*Bhlhe23*	Basic helix-loop-helix family member e23; alias *Dec2*	Transcription factor crucial for neuronal differentiation and retinal development and may also influence circadian rhythm regulation in the brain. Its activity helps shape the development and specialization of certain neuron types [112,113]. Upregulation of Bhlhe23 may increase orexin neuron activity, reducing sleep duration and extending wakefulness. This can lead to impaired cognition, mood disturbances, and increased risk of neuroinflammation and neurodegeneration over time [112].
*Prdm16*	PR/SET domain 16	Transcriptional regulator involved in neurogenesis and the maintenance of neural stem cells [114]. It promotes neuronal differentiation and protects against oxidative stress, supporting normal brain development and function [115]. Dysregulation of Prdm16 in the brain may compromise neurogenesis, increase oxidative stress susceptibility, and negatively affect cognition and long-term brain health, especially under stress or environmental challenge [17,116,117].
*Senp5-ps1*	Small ubiquitin-like modifier (SUMO) protease 5 pseudogene	Pseudogene variant of *Senp5*; non-functional transcript, may regulate their corresponding genes by acting as molecular decoys for microRNAs or generating non-coding RNAs that influence gene expression [118]. Pseudogenes of SUMO proteases may influence neurodegeneration and neuroinflammation by modulating gene expression linked to neuronal survival and cellular stress responses [119]. Changes in their activity could alter susceptibility to neurodegenerative and neuroinflammatory diseases [120].
*Pom121L2*	POM121 transmembrane nucleoporin-like 2	Predicted to encode a nucleoporin-like protein that binds nuclear localization sequences and acts as a structural constituent of the nuclear pore complex. It likely facilitates RNA export and protein import between the nucleus and cytoplasm [121]. Disruption of nuclear pore complex proteins impairs nucleocytoplasmic transport, leading to blood–brain barrier dysfunction, oxidative stress, and chronic neuroinflammation, which collectively increase the risk of neurodegeneration. [122,123]. In a glioma model, Pom121L2 was found to be downregulated in the peritumoral brain region characterized by neuroinflammation and epileptogenic activity [124].
*Chchd4*	Coiled-coil-helix-coiled-coil-helix domain containing 4; alias, Mitochondrial intermembrane space import and assembly protein 40 (*Mia40*)	Essential for mitochondrial protein import; mitochondrial respiration and redox regulation [125], lipid metabolism [126], downregulation suggests energy metabolism deficits in brain [127]. Linked to impaired neuronal energy supply and mitochondrial dysfunction in brain disease [128].

**Table 3 ijms-26-07238-t003:** Lung elastic net predictive gene panel–top 10 contributing genes.

Gene Symbol	Full Gene Name	Biological Function and Disease Relevance
*IgV gene*	Immunoglobulin variable region	Predicted transcript, IgV gene expression in lung injury reflects the activation of the adaptive immune system, particularly B cell recruitment and antibody production [143,144]. While activation of *IgV* may be protective, sustained IgV upregulation is associated with chronic inflammation, tissue remodeling, and fibrosis, as observed in lung diseases such as COPD and interstitial lung disease (ILD) [143,144].
*Kcnq3*	Potassium voltage-gated channel subfamily Q member 3	Potassium channel subunit that regulates membrane potential and sensory neuron excitability in the lung, influencing neuropeptide release, inflammation, and lung injury severity [145]. *Kcnq3* dysregulation can lead to chronic sensory neuron hyperexcitability, promoting airway hypersensitivity, inflammation, and persistent cough. Long-term, this may contribute to airway remodeling and worsening of chronic lung diseases such as asthma or COPD [145].
*Sirpb3*	Signal-regulatory protein beta-3	*Sirpb3* is a pseudogene in the signal regulatory protein (SIRP) family, which includes transmembrane receptors involved in myeloid cell signaling [146]. Though non-coding, *Sirpb3* may regulate innate immune responses through RNA-based mechanisms [147]. Based on its similarity to *Sirpb1* which promotes myeloid activation and inflammation *Sirpb3* may influence immune signaling and contribute to lung inflammatory states [146,147,148].
*Tgfrb1*	Transforming growth factor-beta receptor type 1	Receptor for TGF-β signaling involved in immune regulation and fibrosis; controls lung cell differentiation and matrix production, activation promoting inflammation [149,150]. Persistent *Tgfbr1* activation promotes chronic inflammation, subepithelial fibrosis, smooth muscle thickening, and airway remodeling; impairs epithelial repair and regeneration, leading to fixed airflow obstruction associated with ConB and progressive disease in IPF, COPD, and asthma [151,152].
*Tm4sf19*	Transmembrane 4 L six family member 19	Tetraspanin-like membrane protein that alters lysosomal degradation in macrophages, oxidative stress, and intracellular lipid content and cellular adhesion and immune signaling lysosomal activity in macrophages [153,154]. Dysregulation of Tm4sf19 may impair macrophage lysosomal function and promote oxidative stress, contributing to chronic inflammation, defective lipid clearance, and persistent immune activation processes that underlie airway remodeling and progression of chronic lung diseases such as COPD [153,155].
*St3Gal2*	ST3 beta-galactoside alpha-2,3-sialyltransferase 2	Functions as a sialyltransferase enzyme that catalyzes the addition of sialic acid to glycoproteins and glycolipids, specifically in an α2,3 linkage to galactose residues [156]. This sialylation regulates protein stability, cell signaling, and immune cell interactions. Elevated ST3GAL2 in fibrotic lungs coincides with reduced sialylation, disrupting glycoprotein function and promoting fibrosis [157].
*Tiam1*	T-cell lymphoma invasion and metastasis-inducing protein 1	Cell migration, cytoskeletal remodeling, Rac1 signaling, [64], promotes small cell lung cancer [63]. In the lung, elevated Tiam1 expression has been linked to enhanced invasiveness and metastasis in small cell lung cancer (SCLC). Its dysregulation may also contribute to abnormal epithelial remodeling in chronic lung disease [63].
*Ccl9*	Chemokine (C-C motif) ligand 9	Functions in the lung as a chemokine primarily produced by myeloid cells such as macrophages and dendritic cells. It promotes the recruitment of immune cells through its receptor CCR1, contributing to inflammatory responses and shaping the immune microenvironment [158,159]. It promotes recruitment of dendritic cells and other leukocytes, amplifying inflammatory responses in diseases like asthma [158]. In cancer, especially lung metastasis models, CCL9 induced by TGF-β signaling in myeloid cells enhances tumor cell survival and creates a pro-metastatic lung microenvironment by recruiting immunosuppressive cells [159].
*Foxn4*	Forkhead box N4	Transcription factor expressed in proximal airway epithelial cells during lung development; its loss impairs alveologenesis, leading to enlarged alveoli, thin walls, and poor septation [160,161]. Foxn4 is critical for normal lung development, with its loss leading to impaired alveolar structure and maturation. In lung adenocarcinoma, Foxn4 acts as a tumor suppressor by inhibiting TGF-β signaling and epithelial–mesenchymal transition. Its downregulation promotes tumor progression and inflammation via neutrophil polarization [162].
*Tm4sf1*	Transmembrane 4 L six family member 1	Tetraspanin-like membrane protein that serves as a marker of Wnt-responsive alveolar epithelial progenitor (AEP) cells, contributing to alveolar repair and regeneration after lung injury [66,163]. Supports lung repair by marking Wnt-responsive alveolar progenitor cells involved in regenerating alveolar epithelium after injury [163]. However, in lung cancer particularly NSCLC its overexpression promotes tumor growth, invasion, and epithelial–mesenchymal transition via PI3K/AKT and JAK/STAT signaling. This dual role makes TM4SF1 both a regenerative marker and a potential oncogenic driver [66].

## Data Availability

The data underlying this article will be shared upon reasonablerequest to the corresponding author.

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
