# Peer review of "RNA Sequencing Reveals Inflammatory and Metabolic Changes in the Lung and Brain After Carbon Black and Naphthalene Whole Body Inhalation Exposure in a Rodent Model of Military Burn Pit Exposures"

_ijms, 2025, doi:10.3390/ijms26157238_

Round 1

Reviewer 1 Report

Comments and Suggestions for Authors

This study addresses an important issue regarding inhalation exposures in military personnel due to complex aerosols of burn pit mixtures that can lead to acute and chronic sequelae (aka Deployment Related Respiratory Disease, DRRD). Using a rodent model with napthalene, carbon black, and the combined CBN exposures, the study shows induction of "inflammatory and metabolic gene expression changes in both lung and brain tissues, supporting the utility of this animal model for understanding systemic effects of airborne military toxicants and for identifying potential biomarkers relevant to DRRD and Veteran health." Insofar, as some gene changes were supported by direct measurements of lung, brain, and plasma cytokines and Western blotting, the general conclusions were supported. However, there were issues overall with presentation of the data, the experimental design and methodological details, and the lack of functional readouts that dampened enthusiasm, yet these deficiencies can be addressed (see suggested edits below).

1) Replace bar graphs with individual data points (scatterplots) to improve data transparency and rigor;

2) If necessary, Western blots (WB) should have a MW marker either within the blot or adjacent the protein of interest;

3) Edit the whole manuscript for English grammar and style. The manuscript needs some improvement in language usage throughout. For example, please use the phrase "compared with" in place of "compared to.";

4) Please include a Graphical Abstract as an illustration of all major methods and findings in this manuscript; 

5) The lack of the CB alone exposure group data limits inferences. If the CB exposure was done before, perhaps graphs can include a dashed line to represent the CB only data with an appropriate citation. This will help readers better understand whether there was an additive, synergistic, or other effect.

6) The methods should more clearly indicate what part of the lungs and brain were used for each endpoint. It is unclear from this description: "Briefly, the RNA was isolated from lung and brain tissues by the University of Minnesota Genomics Center using a commercial kit (RNeasy Plus Universal kit, Qiagen, Germantown, MD) according to manufacturer’s instructions." Exactly what portion of the lungs was used and what area of the brain. It's hard to know if the whole brain was used for RNA. What about the cytokines? Were these measured in other brains? This issue needs to be addressed.

Comments on the Quality of English Language

See comments above.

Author Response

We sincerely thank you for your constructive and thoughtful comments on our manuscript, “RNA Sequencing
Reveals Inflammatory and Metabolic Changes in the Lung and Brain After Carbon Black and Naphthalene Whole
Body Inhalation Exposure in a Rodent Model of Military Burn Pit Exposures.” We appreciate the opportunity to
improve our work and believe these revisions have enhanced the clarity and quality of the manuscript.
We thank the reviewers and editors for their time and valuable suggestions, and we look forward to your favorable
consideration.
Should any further clarifications or revisions be required, we would be pleased to address them promptly.
Please find below our detailed responses to the reviewers’ comments and concerns.

REVIEWER 1
1. Replace bar graphs with individual data points (scatterplots) to improve data transparency and rigor.
Response: We appreciate the reviewer’s suggestion to improve data presentation. In response, we have revised our
figures to replace bar graphs with scatterplots displaying individual data points.

We agree that this approach
enhances data transparency and rigor, and we thank the reviewer for this helpful recommendation.

2. Western blots (WB) should have a MW marker either within the blot or adjacent to the protein of interest.
Response: The molecular weight (MW) standard was included in our original western blot images but was previously
labeled only as “kDa.” In response to the reviewer’s comment, we have now clearly labeled the MW markers in the
revised figures to enhance clarity and facilitate interpretation. We appreciate this feedback and believe the revised
presentation improves the quality of our data display.

3. Edit the whole manuscript for English grammar and style.
 The manuscript needs some improvement in language usage throughout. For example, use “compared
with” in place of “compared to.”
Response: We have carefully reviewed all feedback and revised the manuscript accordingly. In particular, we have
thoroughly edited the text to address all concerns related to English grammar, language usage, and style throughout
the manuscript, as requested by Reviewer 1 and the editorial office. Specific attention was given to correcting
grammatical errors, improving sentence clarity, and ensuring appropriate scientific terminology (e.g., using
“compared with” instead of “compared to”).
4. Include a Graphical Abstract as an illustration of all major methods and findings in the manuscript.
Response: We thank the reviewer for this suggestion. In response, we have revised our Graphical Abstract to
include additional details illustrating both the immunological and sequencing methods used in the study. The
graphic is intended to provide a simplified overview of our experimental approach and main findings, rather than
exhaustive methodological or outcome details. We hope that these edits improve the clarity and utility of the figure,
and we appreciate your feedback.
5. The lack of the CB (carbon black) alone exposure group data limits inferences.
If the CB exposure was done before, perhaps graphs can include a dashed line to represent the CB-only data with an
appropriate citation. This will help readers beCer understand whether there was an addiBve, synergisBc, or other effect.

Response: We thank the reviewer for this thoughtful comment. As noted, we have previously published detailed
findings on carbon black (CB)-only exposures (Trembley et al., BMC Res Notes 2022), which established the
immunological and inflammatory effects of CB in this model. However, our primary objective in the present study
was not to dissect the effects of single toxicants, but rather to develop and characterize a more complex, multicomponent
inhalation model that better simulates real-world military burn pit exposures, which are characterized
by mixtures of particulate (CB) and volatile organic (naphthalene) toxicants.
It is important to note that the exposure protocols also differed: rats in the previous CB-only study were exposed to
carbon black for a single 6-hour session, whereas the present study utilized three consecutive days of exposure to
better model subacute, deployment-relevant inhalation scenarios. This difference in exposure time course further
limits the direct comparability of the two datasets and supports our decision not to include the CB-only group in the
current series of analyses. Focusing on complex mixtures rather than single-agent exposures addresses a critical
gap in the field, as human exposures rarely occur to individual chemicals in isolation. By modeling these mixtures,
we aimed to generate mechanistic insights and identify biomarkers that are directly relevant to the types of
exposures experienced by deployed military personnel.
Naphthalene toxicity can vary depending on strain, sex, and exposure protocol, our study design used established
inhalation conditions shown in the literature to induce reproducible, albeit less severe, inflammatory and
transcriptomic responses in rats (Buckpitt et al., Drug Metab Rev 2002). By including this group, we were able to
distinguish the molecular and immune effects attributable to the combined exposures from those produced by
naphthalene alone.
Our study design therefore prioritized comparison of the CBN mixture to naphthalene alone (representing the
volatile fraction) and filtered air controls, in order to assess the unique and combined impacts of multi-component
exposures. While naphthalene-only responses may be more modest or variable, this group serves as a useful
comparator for distinguishing the specific contribution of mixture exposures. We have clarified this rationale in the
revised Methods and Discussion sections and have cited our previous CB-only study for context. We believe this
approach improves the translational relevance of our findings and helps advance the field toward models that better
reflect real-world exposure scenarios.

6. Clarify methodological details in the Methods section, especially regarding tissue sampling.
 Specify what part of the lungs and brain were used for each endpoint, such as for RNA extraction and
cytokine measurements. It is unclear whether whole brain was used, and whether cytokines were
measured in other brains.
Response: We thank the reviewer for highlighting the need for greater clarity regarding tissue sampling. We have
revised the Methods section to specify that whole lung and whole brain tissues were collected and used for both
RNA extraction and cytokine measurements. For each animal, the entire lung and entire brain were snap-frozen,
pulverized to a fine powder under liquid nitrogen, and aliquoted for downstream assays. Both RNA-seq and cytokine
analyses were performed using these homogenized, whole-tissue preparations from the same animals.
We have updated the relevant passages in the Methods to ensure these details are clear, and to indicate that no
regional dissection or pooling from different brains was performed. We appreciate this suggestion, as it improves
the transparency and reproducibility of our methodology.

Reviewer 2 Report

Comments and Suggestions for Authors

This manuscript presents a well-designed and mechanistically informative inhalation toxicology study that models military burn pit exposure using a combination of carbon black nanoparticles and naphthalene in rats. The authors introduce the topic effectively, justify their model with real-world relevance, and provide a clear rationale for their approach. The study is notable for its integration of inflammatory profiling, RNA-seq transcriptomics, and elastic net modeling. They compare their findings to recent literature, acknowledge methodological limitations, and clearly outline how their work complements and improves on existing models. The results support the hypothesis of systemic inflammation and lung–brain axis disruption following short-term inhalation exposure. This is a timely and important contribution to deployment-related health research.

Here are my suggestions/comments/questions:

  1. The study employs a short-term (3-day) exposure window. Could the authors elaborate on the rationale for selecting this time point—was it intended to capture early inflammatory or transcriptomic shifts rather than chronic effects? Expanding briefly on this would clarify the study’s focus on early injury versus long-term changes.
  2. The elastic net model identified a 97-gene lung signature. Do the authors envision this gene panel having potential translational utility, such as in biomarker development or diagnostic screening in clinical or environmental health contexts?
  3. The SenMayo signature showed significant enrichment in brain but not lung. Could the authors comment on whether this reflects a true biological tissue-specific effect, or if it might be due to factors such as timing of sampling, cell-type composition, or limitations in gene set coverage and sensitivity?
  4. Given that bulk RNA-seq was used, do the authors plan to apply single-cell or spatial transcriptomic approaches in the future to better resolve tissue-level responses, clarify the lung–brain axis, and identify cell-type–specific drivers underlying the observed transcriptional changes?

Author Response

Dear Editor and Reviewers,
We sincerely thank you for your constructive and thoughtful comments on our manuscript, “RNA Sequencing
Reveals Inflammatory and Metabolic Changes in the Lung and Brain After Carbon Black and Naphthalene Whole
Body Inhalation Exposure in a Rodent Model of Military Burn Pit Exposures.” We appreciate the opportunity to
improve our work and believe these revisions have enhanced the clarity and quality of the manuscript.
We thank the reviewers and editors for their time and valuable suggestions, and we look forward to your favorable
consideration.
Should any further clarifications or revisions be required, we would be pleased to address them promptly.
Please find below our detailed responses to the reviewers’ comments and concerns.

REVIEWER 2
1. Clarify the rationale for the short-term (3-day) exposure window.

o Was it intended to capture early inflammatory or transcriptomic shifts rather than chronic effects?
Expanding briefly on this would clarify the study’s focus on early injury versus long-term changes.

Response: We thank the reviewer for highlighting the importance of the exposure window. The 3-day protocol was
intentionally selected to model an acute-to-subacute exposure scenario, enabling us to identify early molecular and
inflammatory responses to toxicants, which are critical for understanding the initial triggers of injury and potential
disease progression. This approach builds upon our prior studies (e.g., Trembley et al., 2022) and provides
mechanistic insight into acute initiators of toxicity. We have revised the Discussion section to clarify this rationale
as requested.

2. Discuss translational potential of the elastic net model’s 97-gene lung signature.

o Do the authors envision this gene panel having potential for biomarker development or diagnostic
screening?

Response: We thank the reviewer for highlighting the translational relevance of our findings. We agree that the 97-
gene lung signature identified by our elastic net model represents a promising starting point for future biomarker
development and potential diagnostic applications. While military records can confirm exposure history, this gene
panel offers the potential to objectively assess biological responses and injury, enabling improved diagnostic
accuracy and disease stratification among exposed individuals. Furthermore, it may have utility in therapeutic
testing by serving as a molecular readout of toxicant-induced lung injury and response to intervention.

Our ultimate goal is to refine and validate this gene set ideally reducing it to a smaller subset with robust diagnostic
value in independent preclinical models and, eventually, in human cohorts with deployment-related exposures. We
have expanded our Discussion section to address the translational potential of this gene panel and to outline the
steps needed for future validation, clinical correlation, and development of noninvasive biomarker tools.

3. Comment on SenMayo signature enrichment in brain but not lung.

o Does this reflect a true tissue-specific effect, or could it be due to timing, cell-type composition, or
limitations in gene set coverage?

Response: We thank the reviewer for this insightful question. Our current data support the interpretation that the
brain may be more susceptible than the lung to changes in senescence-related genes following acute inhalation
exposure. The brain’s unique cell-type composition, high metabolic demand, and limited regenerative capacity
may further increase its sensitivity to early senescence pathways.However, we acknowledge that several factors could also contribute to the observed tissue specificity. These
include differences in cell-type composition between lung and brain, potential timing effects (i.e., the window
of sampling relative to the trajectory of the senescence response), and the possibility that the SenMayo gene set
may not capture all relevant senescence programs in the lung. Given the acute nature of our model, we cannot
fully exclude temporal dynamics as a contributing factor, nor can we completely rule out gene set coverage
limitations.
We have expanded our Discussion to address these considerations and to emphasize the need for future studies
using single-cell and/or time-course approaches to better delineate the tissue-specific regulation of
senescence in response to deployment-related inhalation exposures.

4. Discuss future application of single-cell or spatial transcriptomics.

o Given that bulk RNA-seq was used, do the authors plan to apply single-cell or spatial approaches to better
resolve tissue-level responses, clarify the lung–brain axis, and identify cell-type–specific drivers?

Response: We appreciate the reviewer’s suggestion regarding the use of advanced transcriptomic approaches.
We agree that single-cell and spatial transcriptomic analyses would provide critical insights into the cell-type–
specific and spatially resolved responses within lung and brain tissues following exposure. In fact, we plan to
proceed with both single-cell RNA sequencing (scRNA-seq) and spatial transcriptomics in future studies to
better delineate cell-type–specific drivers of injury, clarify inter-tissue communication along the lung–brain axis,
and further characterize the cellular heterogeneity underlying our observed molecular signatures. These
technologies will enable us to capture complex cellular interactions and temporal dynamics that are not
resolvable with bulk RNA-seq. We have added this point to the revised Discussion to highlight our ongoing and
future efforts in this direction.

Round 2

Reviewer 1 Report

Comments and Suggestions for Authors

Reviewer's concerns have been addressed appropriately.